# Global component analysis of errors in three satellite-only global precipitation estimates

Hanqing Chen[a, b, c], Bin Yong[a*], Pierre-Emmanuel Kirstetter[d,e], Leyang Wang[c], Yang Hong[e]

[a]*State Key Laboratory of Hydrology-Water Resources and Hydraulic Engineering, Hohai University, Nanjing 210098, China.*

[b]*School of Geography and Remote Sensing, Guangzhou University, Guangzhou 510006, China.*

[c]*Key Laboratory for Digital Land and Resources of Jiangxi Province, East China University of Technology, Nanchang 330013, China.*

[d]*School of Meteorology, University of Oklahoma, Norman, OK 73072, USA*

[e]*School of Civil Engineering and Environment Sciences, University of Oklahoma, Norman, OK 73019, USA.*

*Corresponding author:* Bin Yong (yongbin@hhu.edu.cn)

**Abstract:** Revealing the error components of satellite-only precipitation products (SPPs) can help algorithm developers and end-users understand their error features and improve retrieval algorithms. Two error decomposition schemes were employed to explore the error components of IMERG-Late, GSMaP-MVK, and PERSIANN-CCS SPPs over different seasons, rainfall intensities, and topography classes. Global maps of the total bias (total mean squared error) and its three (two) independent components are depicted for the first time. Evaluation results obtained between similar regions are discussed and it is found that they cannot be extended to one another region. Hit and/or

false biases are major components of the total bias in most overland regions globally.

The systematic error contributes to less than 20% of the total error in most areas. Large

systematic errors are primarily due to missed precipitation. It is found that the SPPs

show different topographic patterns in terms of systematic and random errors. Notably,

GSMaP-MVK shows a stronger topographic dependency in terms of the four bias

scores. A novel metric namely normalized error component (NEC) was proposed to

isolate the impact of topography on the systematic and random errors. Potential

directions toward improved satellite precipitation retrievals and error adjustment

models were discussed.

**Keywords**: Satellite precipitation; Error component; Systematic error; Rainfall

intensity; Topography

## 1. Introduction

As precipitation is a key input for the hydrological cycle system, accurately estimating

precipitation is of great significances for the study of the global water cycle (Hou et al.,

2014; Kidd et al., 2017; Skofronick-Jackson et al., 2017; Chen et al., 2019a). Traditional

methods depend on rain gauge networks to obtain precise point-scale precipitation

observations (Kidd and Huffman, 2011b). In addition, ground-based radars can provide

remotely-sensed observations for precipitation estimation over a range of

approximately 250 km (Zhang et al., 2016; Chen et al., 2019b). However, these two

methods for estimating precipitation are affected by a number of factors, including local

terrain, weather regimes, environment, and economy. It is challenging to obtain

continuous spatiotemporal precipitation estimates over many regions of the world,

especially over complex terrain and developing countries (Baez-Villanueva et al., 2020).

Satellite-based instruments have the ability to overcome the limitations of rain gauge and ground-based radar networks to provide precipitation estimates covering large areas globally (Kidd et al., 2011a). However, satellite-based precipitation products are

affected by biases and uncertainty, especially over mountains areas (Tian et al., 2010a; Maggioni et al., 2016a; Chen et al., 2020b). Therefore, it is necessary to comprehensively analyze the structure of uncertainty in satellite-based precipitation products, especially those relying on satellite observations only. Over the past 20 years, there is a large body of literature investigating error features of satellite precipitation

products at the global scale (e.g. Yong et al., 2015; Liu et al., 2016; Beck et al., 2017; Chen et al., 2020b) and regional scale (e.g. AghaKouchak et al., 2011; Yong et al., 2010, 2013, 2016; Takido et al., 2016; Tan et al., 2017; Prakash et al., 2018; O and Kirstetter, 2018; Gebregiorgis et al., 2018; Beck et al., 2019; Chen et al., 2019b). These studies have provided a great deal of valuable information for algorithm developers and end-

users. Most studies use the mean error to analyze the error features of SPPs, which could be misleading because the mean error averages different error components. In some cases, the error components compensate each other and generate lower mean error values than the absolute values of the individual components (Chen et al., 2019b).

Tian et al. (2009) proposed an error decomposition scheme to separate the total bias into three independent components (i.e., hit bias, miss bias and false bias). To date,

several evaluation studies investigate major bias components of satellite precipitation products over several regions, such as Mainland China (Yong et al., 2016; Xu et al., 2016; Su et al., 2018; Chen et al., 2020b), United States (Tian et al., 2009), Central Asia (Guo et al., 2017). AghaKouchack et al. (2012) used an error decomposition technique proposed by Willmott, (1981) to separate the total mean squared error into the systematic and random errors for evaluating three satellite precipitation products (i.e., CMORPH, PERSIANN, and real-time TMPA) over the conterminous United States (CONUS). Maggioni et al. (2016b) further investigated the systematic errors of TMPA products over CONUS. However, these studies were concentrated on limited regions and did not provide a global focus. Importantly, the question of transferability of regional evaluation results to different but similar areas still needs to be investigated. It has implications in terms of improving the performance of SPPs in regions of the world where no ground observations are available. Besides, investigation is still need on the major component of the total bias that produces the largest systematic errors.

Topography is a crucial factor that influences satellite precipitation retrievals (Tapiador et al., 2012; Xu et al., 2017; Chen et al., 2019b). While several studies have strived to investigate the total bias of satellite precipitation retrievals over various topography (e.g. Takido et al., 2016; Guo et al., 2017; Xu et al., 2017; Chen et al., 2019b), error components remain underexplored. In particular, the literature that investigates the potential link between systematic (random) error and terrain features is lacking, which limits the characterization of satellite precipitation error. Furthermore, previous studies

used the mean elevation as a way to describe the underlying terrain. As the mean

elevation often does not objectively represent the complexity of topography, a more

relevant metric is needed.

Precipitation intensity is another important impact factor driving errors of satellite

precipitation estimates (Tian et al., 2009; Kirstetter et al., 2013; Chen et al., 2013; Chen

et al., 2020b). Previous efforts found that satellite precipitation products tend to

overestimate light rainfall and underestimate heavy rainfall (Tian et al., 2009; Kirstetter

et al., 2013; Chen et al., 2013). Tian et al. (2009) investigated the major components of

the total bias in six SPPs (e.g. AFWA, TMPA suite, CMORPH, PERSIANN, and NRL)

for different rainfall intensities, and Maggioni et al. (2016b) quantified the relationship

between the systematic (random) error of TMPA products and rainfall intensity, while

Kirstetter et al. (2018) revealed the relationship between the systematic (random) error

in PERSIANN-CCS and precipitation intensity. Nevertheless, the potential links

between the systematic (random) error of IMERG-Late and GSMaP-MVK and

precipitation intensity are still absent.


The objective of this study is four-fold: (1) to reveal the major components of errors

(including total bias and total mean squared error) for three SPPs including IMERG

Late run (IMERG-Late), GSMaP Microwave-IR Combined Product (GSMaP-MVK),

and PERSIANN-CCS for four seasons across global land areas; (2) to investigate the

potential for the transferability of the regional assessment results to other similar

regions; (3) to investigate what are the factors causing large systematic errors; (4) to inform users and algorithm developers on how to improve these satellite precipitation products.

## 2. Study area, datasets and methodology

### 2.1 Study area

The study area covers the global lands (60°N/S). Fig.1a shows the topographic relief in terms of standard deviation of elevation (SDE; more information in section 2.3.3). SDE was introduced to better describe the topography at each grid cell. The variability of topography highlights complex terrain areas that include the western CONUS, the Andean mountains, southern Europe, Turkey, Iran, Afghanistan, the Tibetan Plateau (TP), most humid regions in mainland China, and Japan, etc. The global land areas can be divided into four climate regions namely humid regions (average annual precipitation (AAP) > 800mm/yr), semi-humid regions (AAP between 400 – 800 mm/yr), semi-arid regions (AAP between 200 – 400 mm/yr), and arid regions (AAP < 200 mm/yr), as shown in Fig. 1b. The geographical distribution of the climate regions can be found in Fig.1c.

### 2.2 Datasets

### 2.1.1 Reference products

Three high-accuracy rain gauge data sets are employed as the references. The Climate Precipitation Center unified (CPCU) data are used as the benchmark over the global land areas except for mainland China. CPCU produces continuous daily precipitation

at 0.5º spatial resolution using optimal interpolation (OI) based on > 17,000 gauges (Xie et al., 2007; Chen et al., 2008). For the benchmark over mainland China, China Gauge-based Daily Precipitation Analysis (CGDPA) data are used. This dataset, with a 0.25º spatial and daily temporal resolution, is developed based on ~ 2400 rain gauges using the OI method. Assessment results indicate that this ground-based precipitation dataset outperforms CPCU and the East Asia gauge analysis (EA_Gauge; Xie et al., 2007) over mainland China (Shen and Xiong, 2016). To alleviate the effects of reference inconsistency on the analysis, the error scores of SPPs over mainland China and the rest of the world are calculated separately and subsequently merged. The detailed procedure is provided in supplementary material. Regarding the component analysis of SPP errors in different topographies, high-accuracy and high spatiotemporal resolution ground observations (hourly and 0.1°) derived from 25,982 rain gauges (Shen et al., 2014) are used as the benchmark. The spatial distribution of the rain gauge can be found in Chen et al. (2019b) and Chen et al. (2020b). However, this product has large uncertainties in cold seasons due to freezing weather. The analysis was executed at a 0.1° spatial resolution to avoid smoothing the topography features as much as possible. In this study, only the pixels with at least one rain gauge are considered. The spatial distribution of rain gauges including CPCU and CGDPA is shown in Fig. 1d.

**2.1.2 Satellite-only precipitation products**

The main focus of this study is to analyze the error components of the three SPPs (i.e. IMERG-Late V6, GSMaP-MVK V7, and PERSIANN-CCS). The corresponding gauge-adjusted satellite precipitation products (IMERG Final run, gauge-adjusted

GSMaP, and PERSIANN Climate Data Record) that merge ground-based rain gauge

and satellite observations were not used in this study to ensure an objective assessment

with independent benchmarks (e.g. the gauge-adjusted GSMaP ingests CPCU data).

Note that other satellite-only SPPs are not included in this study, either because they

are not released to the public (e.g. CMORPH) or they have been decommissioned (e.g.

real-time TMPA; https://gpm.nasa.gov/). A 5-year period (2015-2019) was chosen to

investigate SPPs error components. Table 1 provides general information on the three

SPPs and more detailed information about their production processes can be found in

Chen et al. (2020b).

To be consistent with CPCU data (0.5º, daily), all SPPs are resampled to the same 0.5º

spatial resolution and aggregated to daily temporal resolution.

## 2.3 Methodology

### 2.3.1 Error decomposition technique

Tian et al. (2009) proposed an error decomposition scheme to separate the total bias

(TB) into hit bias (HB), miss bias (MB), and false bias (FB). This technique is effective

at identifying the major error components of the total bias, and can provide valuable

information to customize retrieval algorithms and mitigate errors. The four bias scores

can be defined as follows (Tian et al., 2009):

$$TB = \frac{\sum(S-G)}{\sum G} \times 100\% \qquad (1)$$

$$HB = \frac{\sum(S_H - G_H)}{\sum G} \times 100\% \qquad (2)$$

$$MB = \frac{-\sum G_M}{\sum G} \times 100\% \qquad (3)$$

$$FB = \frac{\sum S_F}{\sum G} \times 100\% \tag{4}$$

$$TB = HB + MB + FB \tag{5}$$

where $S$ and $G$ are the precipitation measured by satellite and rain gauge, respectively; $S_H$ and $G_H$ are the precipitation estimates of hit rainfall events for satellite and rain gauge, respectively; $G_M$ denotes the precipitation missed by satellite in miss rainfall events; $S_F$ indicates the precipitation measured by satellite in false rainfall events.

Another error decomposition technique decomposes the total mean squared error into systematic and random error components. This strategy was used to separate numerical weather prediction models into systematic and random errors by Willmott, (1981). Subsequently, AghaKouchack et al. (2012) employed this technique to investigate the systematic and random errors of three satellite precipitation products (i.e., CMORPH, PERSIANN, and real-time TMPA) over CONUS. This error decomposition method can be defined as follows (Willmott, 1981; AghaKouchack et al., 2012):

$$\frac{1}{n}\left(\sum_{i=1}^{n}(S-G)^2\right) = \frac{1}{n}\left(\sum_{i=1}^{n}(\hat{S}-G)^2\right) + \frac{1}{n}\left(\sum_{i=1}^{n}(S-\hat{S})^2\right) \tag{6}$$

$$E_S = 100\% \times \left(\sum_{i=1}^{n}(\hat{S}-G)^2\right)/(\sum_{i=1}^{n}(S-G)^2) \tag{7}$$

$$E_R = 100\% \times \left(\sum_{i=1}^{n}(S-\hat{S})^2\right)/(\sum_{i=1}^{n}(S-G)^2) \tag{8}$$

$$\hat{S} = a \times G + b \tag{9}$$

where $E_S$ and $E_R$ represent the systematic and random components of error, respectively; $a$ and $b$ are slope and intercept, respectively, and they can be computed by using least square method. Note that the systematic error component ($E_S$) plus random error component ($E_R$) add up 100%.

**2.3.2 Normalized error component**

The systematic and random errors of SPPs are impacted by several key impact factors, such as season and rainfall intensity (AghaKouchack et al., 2012; Maggioni et al., 2016b; Kirstetter et al., 2013, 2018). To isolate the influence of topography on the systematic and random errors, a novel metric called the normalized error component (NEC) is proposed that filters the impact of precipitation intensity. This metric is defined as follows:

$$NEC = \left(\sum_{i=1}^{n}(\hat{S} - G)^2\right) / \left((\sum_{i=1}^{n}(S - G)^2) \times \bar{G}\right) \tag{10}$$

where $\bar{G}$ indicates the mean value of ground-based observations for each topographic class. Note that the relationship between NEC and topography degenerates into the relationship between systematic error and topography when the mean precipitation (i.e., $\bar{G}$) of all topography classes is close. Thus, the NEC metric works only when the rainfall intensities are significantly different across all topographic categories.

**2.3.3 Index of topography complexity**

To better describe the variability of topography for each grid cell, the standard deviation of elevation (SDE) is proposed instead of average elevation. The larger the SDE value, the higher the terrain gradients within the grid cell, reflecting steeper topography. The SDE formula is defined as follows:

$$\begin{cases} \bar{E} = \frac{1}{n}\sum E_i \\ SDE = \sqrt{\frac{1}{n}\sum(E_i - \bar{E})^2} \end{cases} \tag{11}$$

where $\bar{E}$ indicates the mean value of elevation for each pixel; $E_i$ denotes $i$th elevation value of each grid cell; $n$ represents the elevation sample size of each pixel. The global

map of SDE is shown in Fig. 1a. The SPPs errors and their components are computed

as functions of SDE under different topographies. The relationship between error and

precipitation intensity is similarly established for various rainfall rate classes to

investigate the impact of rainfall intensity.

## 3. Results

### 3.1 Global view of error components

### 3.1.1 Spatial analysis of error components over different seasons

As the seasonal dependency of SPPs errors has been documented (Tian et al., 2009;

Chen et al., 2019), the analysis first focuses on different seasons. The following

seasonal division scheme was implemented: (1) Spring (Mar – May, hereafter refer to

as MAM); (2) Summer (Jun – Aug, hereafter refer to as JJA); (3) Fall (Sep – Nov,

hereafter refer to as SON); (4) Winter (Dec – Feb, hereafter refer to as DJF).

For MAM season (Fig. 2), the majority of SPPs share considerable similarities in total

bias. IMERG-Late and GSMaP-MVK seriously overestimate precipitation (total bias >

100%) over most regions of the globe, such as the humid regions of CONUS, Mexico,

Europe, India, and the semi-humid areas of China (see Figs. 2a, e). Yet, the total biases

come from different error components in different areas. The total biases of IMERG

and GSMaP are mainly dominated by hit component in CONUS and Europe, while the

hit and false errors dominate in Mexico, India and mainland China. As for PERSIANN-

CCS, its larger total biases primarily occur in CONUS (except for its humid regions),

Mexico, Brazil, and most of land areas in Asia (except for humid regions in China). Correspondingly, these larger total biases originate from hit and false components over CONUS and Brazil, while they are dominated by false errors over Asia and Mexico.

Regarding the JJA season (Fig. 3), the three SPPs exhibit large overestimations over most of CONUS (total bias > 80%), which primarily owe to the hit error component for IMERG and GSMaP. As for northwest Mexico, the precipitation is evidently overestimated with hit bias by the three SPPs. Regarding Europe, the hit error is also the major error component for IMERG and GSMaP. PERSIANN-CCS underestimates
precipitation over most regions of Europe, due to miss errors and hit errors. In addition, the SPPs show relatively better performance in mainland China during the JJA season (Summer), with a lower total bias of ± 20%.

As for the SON season (Fig. 4), SPPs share considerable differences in the error features.
Over most regions of CONUS, IMERG displays relatively low overestimation and underestimation. However, GSMaP seriously overestimates precipitation over CONUS due to its larger hit biases. PERSIANN-CCS overestimates precipitation in arid and semi-arid regions of CONUS, which is attributed to hit and false components, while it underestimates precipitation over southeastern regions due to the hit error component.
Over most regions of China, IMERG shows slight underestimation or overestimation. GSMaP and PERSIANN-CCS sorely underestimate (overestimate) precipitation in humid areas (except for humid regions) due to a larger proportion of missed

precipitation (false error component). On the other hand, all SPPs have in common a feature that their total biases are very similar to their hit error components over Mexico, Brazil, Europe, and India, because missed precipitation and false bias cancel one another.

Global maps of the four bias metrics for the three SPPs over the DJF season are shown in Fig. 5. One can notice that the three SPPs display an obvious similarity in error characteristics over Brazil and Australia. Again their total biases are very similar to their respective hit error component. On the other hand, differences in the four bias scores exist in CONUS, Europe, and China among the SPPs. Different retrieval algorithms and input sources used in different satellite products may explain these differences.

The summary of the total bias and its major error components in the main regions of the world is listed for the three SPPs in supplementary materials (Table S1). Overall, the global maps of the total bias and its three independent error components indicate that hit and/or false error components are the major contributor of the total bias.

**3.1.2 Spatial analysis of the systematic error over different seasons**

The three SPPs share considerable similarities in the global maps of systematic errors for the four seasons over most global land areas. The systematic error is less than 20% of the total mean squared error as shown in Fig. 6. It means that the random error is the leading error component of the total mean squared error in most regions. As discussed

in the previous section, the SPPs show an evident seasonal dependency in several regions such as CONUS, China, and Russia. IMERG has relatively larger systematic errors exceeding 80% in the semi-humid and semi-arid regions of CONUS during DJF. Similarly, the systematic errors of IMERG are very large (over 90%) over mainland China (except for humid regions) during DJF. For GSMaP, the seasonal variations mainly occur in mainland China and Russia. Systematic errors are larger in DJF than other seasons over Russia. Meanwhile, it cannot be ignored that GSMaP exhibits large proportions of systematic error during SON in southwest China. Finally, the PERSIANN-CCS systematic error displays an obvious seasonality in mainland China, Europe, and eastern CONUS and it is relatively larger in over mainland China.

**3.2 Error components dependency with precipitation intensities**

The three bias scores (i.e., total bias, hit bias, and miss bias) of the three SPPs are shown in Fig. 7 for different rainfall intensities. Note that the false error component does not exist because the reference precipitation is positive. In general, the SPPs show a high degree of consistency in the three bias scores for different precipitation intensity classes. The hit bias is the major error component in most classes.

The variations of systematic error with six rainfall intensities are depicted in Fig. 8. Each SPP shows unique variations of systematic errors with increasing precipitation intensity. One can see that IMERG and GSMaP have less systematic error (close to 0%) for intensities less than 40mm/day. In contrast, PERSIANN-CCS systematic error shows a strong and increasing relation with rainfall intensity. Additionally, all SPPs

have relatively larger proportions of systematic error and underestimate the precipitation volume in rainfall events where rainfall intensities are over 40 mm/day (see Fig. 7). This underestimation in such heavy rainfall events certainly results in large systematic errors.

## 3.3 Error components for different topographies

In this section, the analysis is performed at a finer spatiotemporal resolution (0.1°, hourly). Additionally, the humid regions of mainland China during JJA are chosen to explore the major components of the total bias and total mean squared error under different topography categories, to exclude interferences with other factors (e.g. climate and season). Observations from 25,982 rain gauges are used as the benchmark to ensure the reliability of the error results (Chen et al., 2020a). The gauge number in each topography category is displayed in Fig. 9.

The four bias metrics are shown in Fig. 10 for different terrain classes. The SPPs share similarities in the variations of the four bias scores with terrain. The miss biases decrease with increasing SDE. Also, miss (false) bias mainly contributes to the underestimated (overestimated) precipitation. The performance of all SPPs is impacted by topography to different extents. Relatively, the four GSMaP-MVK biases metrics display more significant topographic dependency, especially in terms of false and hit error components. A possible explanation is that the orographic/non-orographic rainfall classification scheme used in the GSMaP retrieval tends to overestimate precipitation in the hit orographic rainfall events and is affected by false-positive in orographic

rainfall (Yamamoto and Shige, 2015).

330

Fig. 11a shows the proportions of the systematic error for the three SPPs under eleven terrain SDE categories. Obviously, the systematic errors of all SPPs display a strong topographic dependency, yet SPPs exhibit distinct dependency patterns. For IMERG-Late, the proportion of the systematic error increases with SDE. GSMaP-MVK and

335 PERSIANN-CCS display similar trends in that the proportions of systematic error increases with topography complexity when SDE is below 300 m then decreases with SDE above 300 m. Compared to other SPPs, PERSIANN-CCS has relatively larger systematic errors in all topographic categories. Nevertheless, the results shown in Fig. 11a may be impacted by rainfall intensity, making any inference on the influence of

340 topography challenging. This issue is substantially investigated in section 4.2.

## 4. Discussion

### 4.1 Potential for transferability of the regional assessment to other areas

While there are numerous evaluations of mainstream SPPs over regions of the Earth's land, such as CONUS, Europe, India, China, many other regions lack the needed ground

345 networks to assess SPPs such as Africa, central Australia, Mongolia, etc. It raises the important question of extrapolating evaluation results to other similar areas where no ground observations are available. The transferability of evaluation results to other regions needs to be investigated.

The comparisons of total bias between the Chinese Fujian (FJ) and Zhejiang (ZJ)

provinces are regarded as a representative example for such analysis. The two areas are

located in the humid regions of mainland China. They are dominated by the same

monsoon climate and have similar topography. The spatial distribution of the total bias

for each SPP shows large differences between FJ and ZJ provinces, as shown in Fig. 12.

Besides, the total biases in FJ (or ZJ) show evident differences for each SPP. It appears

as if the evaluation results between similar areas may not be extended to one another.

Chen et al. (2019b) found that large differences in performance exist between various

sensors onboard different satellites, that may be related with sampling frequency (see

Fig. 2 in Chen et al., 2019b). Also, that may be also caused by other factors (e.g.

characteristics of precipitation regimes, such as precipitation types) not captured by

satellites or the reference. Future evaluation efforts should focus on ungauged regions

and explore novel methods that do not depend on ground observations.

## 4.2 Impact of topography on the systematic error

In section 3.2, the results indicated that systematic errors are related with rainfall

intensity. Although the analysis is performed in the humid regions of mainland China

during the summer season to alleviate interferences of climate and seasonal factors on

the systematic error, discrepancies in precipitation intensity that affect the proportions

of SPP systematic error are expected between different topography categories. The NEC

metric is designed to exclude the impact of the precipitation intensity on the systematic

error. It is used to extract the influence of topography on the systematic error.

Fig. 11b shows the variations of NEC values for the three SPPs as a function of SDE for the summer season over the humid regions of mainland China. It is obvious that the relationship between the NEC and SDE is highly similar to that between the systematic error and SDE. The similarity of the two relationships can be explained as the mean precipitation (i.e., $\bar{G}$, see equation (10)) of all SDEs is close around 0.24 mm/h. The relationship between NEC and SDE degenerates into the relationship between systematic error and SDE once the mean precipitation at all topographic classes is close. According to the results shown in Fig. 11b, the impacts of topography on the systematic error for IMERG increases with terrain complexity. For GSMaP and PERSIANN-CCS, the systematic error increases (decreases) with topographic complexity when SDE is less than (above) 300 m.

## 4.3 What are the factors causing larger systematic error?

In general, the proportions of systematic error for the three evaluated SPPs are below 20% for all four seasons and over most of the global land areas. However, it cannot be ignored that these SPPs have larger systematic errors in several regions, such as parts of CONUS, China and Russia (see Fig. 6). In addition, SPPs in these areas with larger systematic errors have always relatively larger miss biases (see Figs. 2-6), implying that miss bias tends to produce larger systematic errors relative to hit and false biases, according to the definition of systematic and random errors (see equations (7-8)).

## 4.4 Potential directions of the improvement in satellite retrieval algorithms and error adjustment models

The results in this study suggest that the quality of the evaluated SPPs have significant

room for further improvement. Several recent studies attempted to reduce the errors in

satellite precipitation retrievals by only considering seasonal, rainfall intensity, and/or

topographic factors into their error adjustment models or blending algorithms (e.g. Tian

et al., 2010b; Hashemi et al., 2017; Bhuiyan et al., 2018; Le et al., 2018; Choubin et al.,

2019; Shen et al., 2019; Baez-Villanueva et al., 2020). In practice, the errors show

significant regional features (at least for the three evaluated SPPs). The impact of

several key factors (i.e., topography, season, climate, and rainfall intensity) is very

significant, suggesting that incorporating all four factors (i.e., topography, season,

climate region/different areas, and rainfall intensity) into error adjustment models and

blending algorithms is expected to further reduce the errors of satellite precipitation

estimates.


Second, the global maps of total bias (total mean squared error) and its three (two)

components indicate that the hit and/or false errors are the major contributor of the total

bias. The random error is found to be the major component of the total mean squared

error. Consequently, the satellite retrieval algorithms and error adjustment algorithms

should focus on reducing the hit and false biases in these SPPs over most regions of the

world.

Finally, the findings of this study are relevant to the improvement of gauge-adjusted

versions of the satellite retrievals (e.g. GSMaP-Gauge blends GSMaP-MVK and CPCU

data).

## 5. Conclusion

This paper investigates major error components of the total error for three SPPs (i.e., IMERG-Late, GSMaP-MVK, and PERSIANN-CCS) over different seasons, rainfall intensities, and various topography. The major conclusions are summarized as follows:

1. This study is the first to depict global maps of the total bias (total mean squared error) and its three (two) independent components for three SPPs over four seasons. The errors are found to have remarkably regional features, and the evaluation results show limited transferability from one region to another. This can be attributed to differences in satellite sampling between areas and may be

also caused by other factors (e.g. characteristics of precipitation regimes, such as precipitation types), which cannot be captured by satellites or the reference. This finding highlights the need for assessing satellite precipitation products over various regions of the world. Future efforts should focus on areas with lack of evaluation and on investigating novel evaluation techniques that do not

rely on ground-based observations.

2. Hit and/or false errors are the major components of the total bias for the three SPPs over most areas of the world (see Table S1). The proportions of the systematic error are below 20% and display a strong seasonality in several regions, such as CONUS, China, and Russia. It appears that missed

precipitation is a decisive factor producing large systematic errors. The evaluation results indicate that the satellite retrieval algorithms and error adjustment algorithms should focus on reducing the hit and false biases in these

20 / 51

SPPs over most regions of the world.

3. All SPPs exhibit a high degree of consistency in the three bias scores (i.e., total bias, hit bias, and miss bias) under different rainfall intensities. Their total biases came primarily from the hit error component. Each SPP displays a specific relationship between the systematic error and precipitation intensity. All SPPs have relatively larger systematic error in rainfall events with intensity exceeding 40 mm/day.

4. All SPPs share considerable similarities in terms of the four bias metrics (i.e., total bias, hit bias, miss bias, and false bias) over most SDE classes. Relatively, the four bias scores of GSMaP have a stronger topographic dependency, especially for false bias and hit bias. The three SPPs exhibit distinctly various topographic dependency patterns in systematic error. The NEC metric was proposed to isolate the influence of topography on the systematic error. It is found that the relationship between NEC and topography degenerates into the relationship between systematic error and topography, primarily due to mean precipitation (i.e., $\bar{G}$, see equation (10)) of ~ 0.24 mm/h in all terrain categories.

The new findings reported in this paper will be useful to improve satellite precipitation retrieval algorithms and error adjustment models, as well as the potential applications of these products.

## Code/Data availability

The IMERG suite data can be obtained from https://pmm.nasa.gov/data-access/downloads/gpm; the GSMaP suite data are obtained from ftp://rainmap:Niskur+1404@hokusai.eorc.jaxa.jp/; the ground-based data in mainland China can be downloaded from http://data.cma.cn; the CPCU data can be downloaded from ftp://ftp.cpc.ncep.noaa.gov/precip/. MATLAB codes used in this study are available by contacting the first author (hanqingchen1007@163.com).

## Author contributions

**Hanqing Chen**: Conceptualization, Methodology, Software, Formal analysis, Writing, Funding Acquisition. **Bin Yong**: Writing – Review & editing, Project Administration, Funding Acquisition. **Pierre-Emmanuel Kirstetter:** Writing – Review & editing. **Leyang Wang**: Writing – Review & editing, Methodology. **Yang Hong**: Conceptualization.

## Competing interests

The authors declare that they have no conflict of interest.

## Acknowledgments

We would like to express our most sincere thanks to the Editor and two anonymous reviewers for their efforts in improving the quality of this paper. This work was supported by National Key Research and Development Program of China (No. 2018YFA0605402) and National Natural Science Foundation of China (No. 51979073).

In addition, this work is partially supported by Key Laboratory for Digital Land and Resources of Jiangxi Province, East China University of Technology (No. DLLJ201907).

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

## Figure and Table Captions

**Fig. 1.** (a) Global map of topography; (b) mean annual precipitation of the global land from 1891 to 2018 (128 years) based on the Global Precipitation Climatology Centre (GPCC) monthly gauge analysis; (c) global land is divided into four climate areas (i.e., humid, semi-humid, semi-arid, arid); (d) the spatial maps of rain gauges used in Climate Precipitation Center unified (CPCU) data and China Gauge-based Daily Precipitation Analysis (CGDPA) data.

**Fig. 2.** Global maps of the total bias and its three independent components for the three satellite-only precipitation global precipitation estimates (i.e., IMERG-Late, GSMaP-MVK, and PERSIANN-CCS) at a 0.5° spatial and daily temporal resolution in the MAM season (Mar-May) across global land for the period from 2015 to 2019 (5 years).

**Fig. 3.** Global maps of the total bias and its three independent components for the three satellite-only precipitation global precipitation estimates (i.e., IMERG-Late, GSMaP-MVK, and PERSIANN-CCS) at a 0.5° spatial and daily temporal resolution in the JJA season (Jun-Aug) across global land for the period from 2015 to 2019 (5 years).

**Fig. 4.** Global maps of the total bias and its three independent components for the three satellite-only precipitation global precipitation estimates (i.e., IMERG-Late, GSMaP-MVK, and PERSIANN-CCS) at a 0.5° spatial and daily temporal resolution in the SON season (Sep-Nov) across global land for the period from 2015 to 2019 (5 years).

**Fig. 5.** Global maps of the total bias and its three independent components for the three satellite-only precipitation global precipitation estimates (i.e., IMERG-Late, GSMaP-

MVK, and PERSIANN-CCS) at a 0.5° spatial and daily temporal resolution in the DJF

season (Dec-Feb) across global land for the period from 2015 to 2019 (5 years).

**Fig. 6.** Global maps of the systematic errors for the three satellite-only global

precipitation estimates (i.e., IMERG-Late, GSMaP-MVK, and PERSIANN-CCS) at a

0.5° spatial and daily temporal resolution in four seasons (i.e., MAM, JJA, SON, and

DJF) over global land for the period from 2015 to 2019 (5 years): (a-d) IMERG-Late,

(e-h) GSMaP-MVK, (i-l) PERSIANN-CCS.

**Fig. 7.** Three bias scores (i.e., total bias, hit bias, and miss bias) of the three satellite-

only global precipitation estimates (i.e., IMERG-Late, GSMaP-MVK, and

PERSIANN-CCS) at a 0.5° spatial and daily temporal resolution under different rainfall

intensities. Note that the precipitation intensity categories are from references.

**Fig. 8.** Systematic errors of the three satellite-only global precipitation estimates (i.e.,

IMERG-Late, GSMaP-MVK, and PERSIANN-CCS) at a 0.5° spatial and daily

temporal resolution under different rainfall intensities. Note that the precipitation

intensity categories are from references.

**Fig. 9.** Gauge number for each topography class.

**Fig. 10.** Four bias scores (i.e., total bias, hit bias, miss bias, and false bias) of the three

satellite-only global precipitation estimates (i.e., IMERG-Late, GSMaP-MVK, and

PERSIANN-CCS) under different terrains. Note that the analysis executed at a 0.1°

spatial and hourly temporal resolution in the humid regions of mainland China over JJA

(summer) season for the period from 2015 to 2019 (5 years).

**Fig. 11.** (a) Systematic errors for the three satellite-only global precipitation estimates (i.e., IMERG-Late, GSMaP-MVK, and PERSIANN-CCS) under different topographies; (b) the variations of normalized error component (NEC) for the three satellite-only global precipitation estimates with increasing terrain complexity. Note that the analysis executed at a 0.1° spatial and hourly temporal resolution in the humid regions of mainland China over JJA (summer) season for the period from 2015 to 2019 (5 years).

**Fig. 12.** Spatial maps of the total biases of the three SPPs for four seasons over the Fujian (FJ) and Zhejiang (ZJ) provinces, respectively.

**Table 1** The information about three satellite-only global precipitation estimates used in this study.

**Figures**

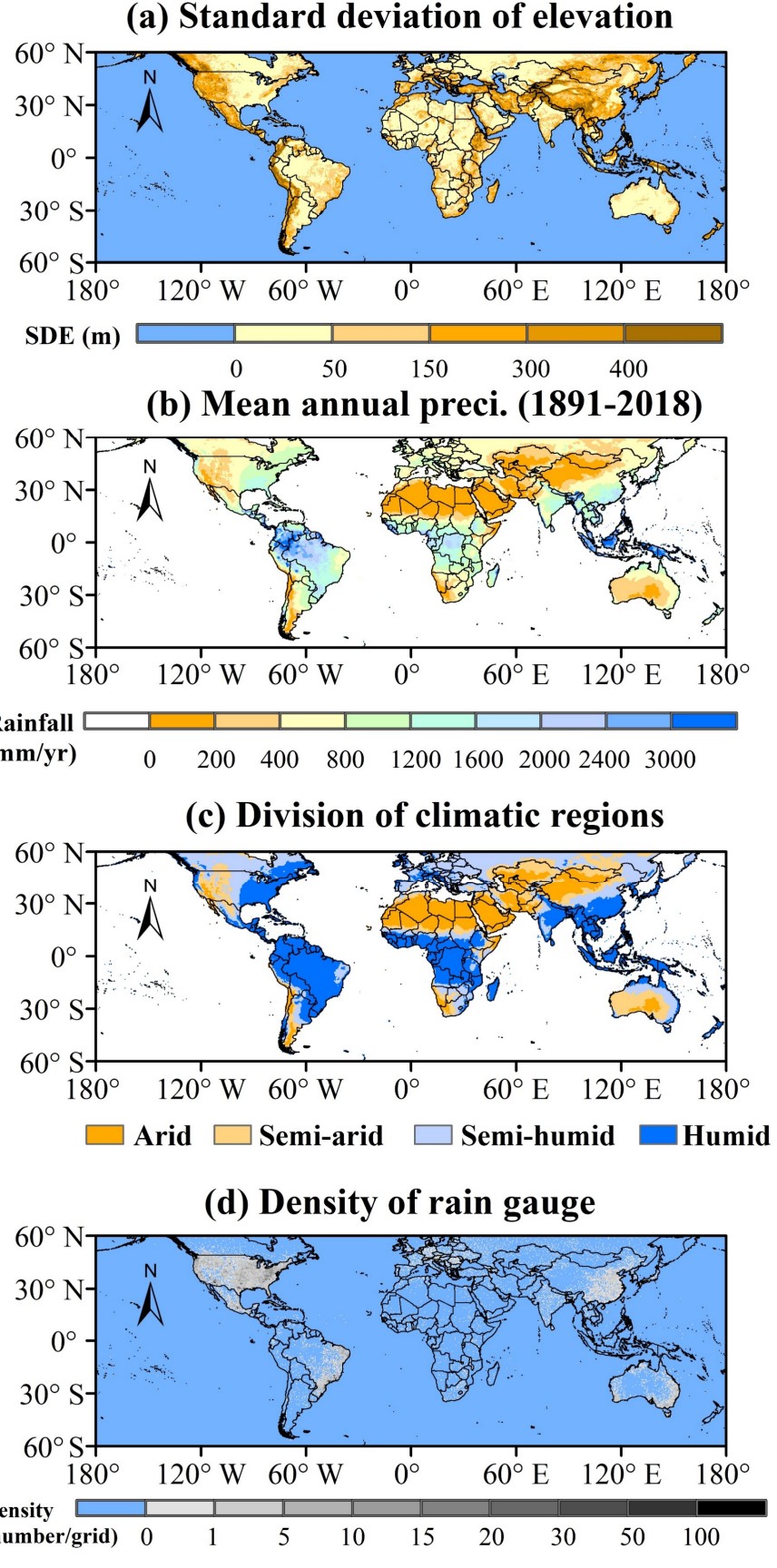

**Fig. 1.** (a) Global map of topography; (b) mean annual precipitation of the global land from 1891 to 2018 (128 years) based on the Global Precipitation Climatology Centre (GPCC) monthly gauge analysis; (c) global land is divided into four climate areas (i.e., humid, semi-humid, semi-arid, arid); (d) the spatial maps of rain gauges used in Climate Precipitation Center unified (CPCU) data and China Gauge-based Daily Precipitation

Analysis (CGDPA) data.

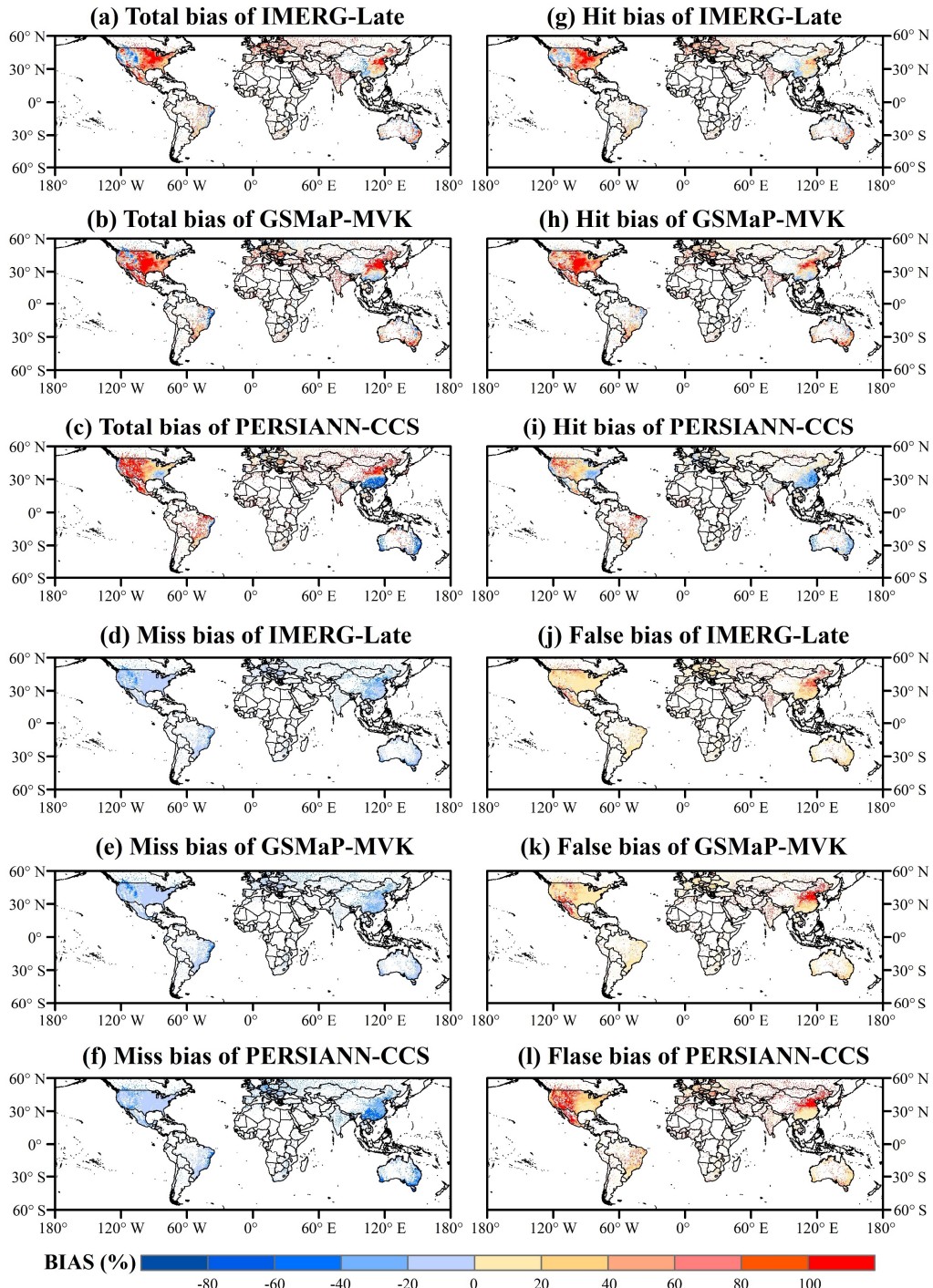

**Fig. 2.** Global maps of the total bias and its three independent components for the three satellite-only precipitation global precipitation estimates (i.e., IMERG-Late, GSMaP-MVK, and PERSIANN-CCS) at a 0.5° spatial and daily temporal resolution in the MAM season (Mar – May) across global land for the period from 2015 to 2019 (5 years).

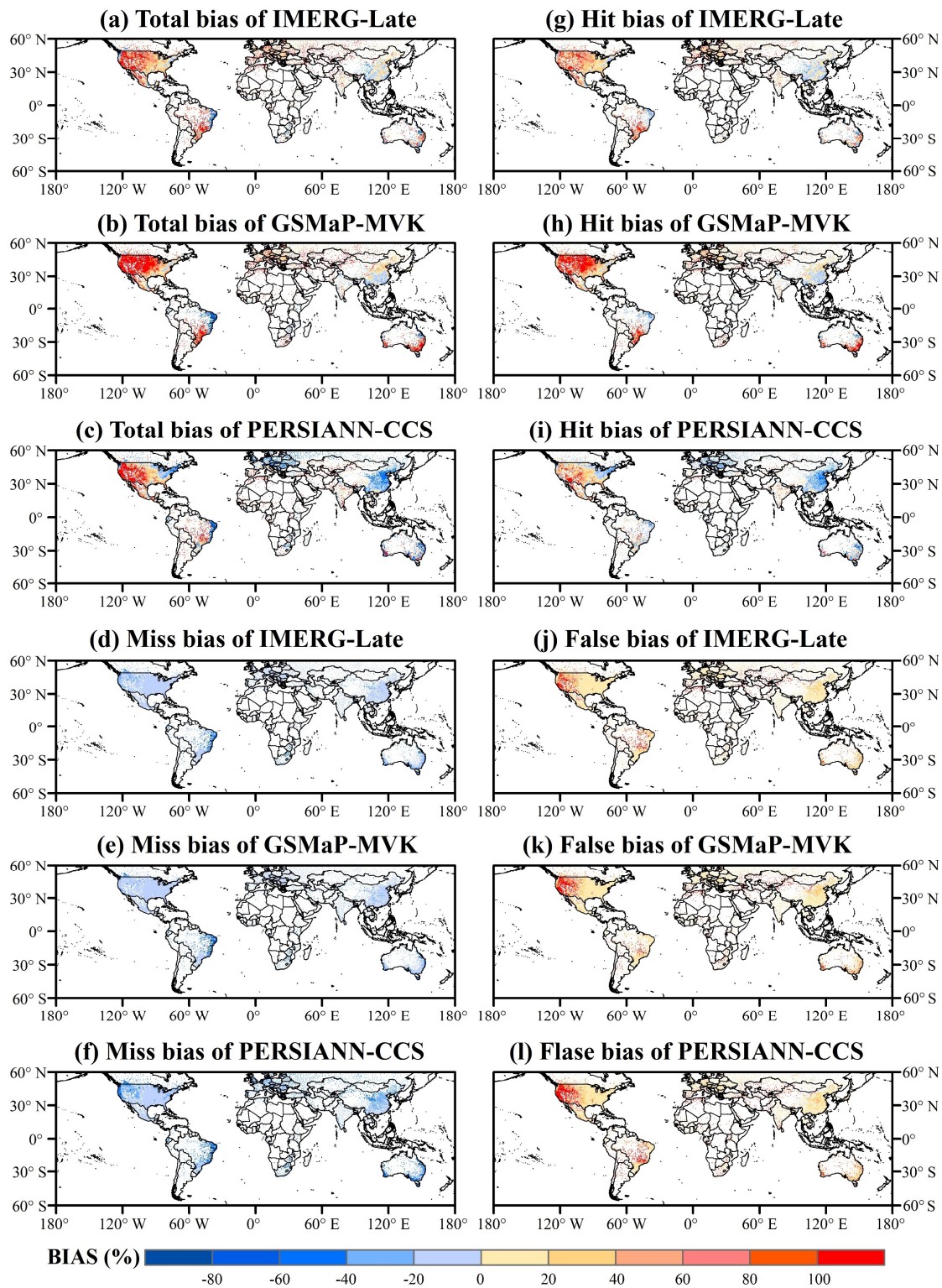

**Fig. 3.** Global maps of the total bias and its three independent components for the three

satellite-only precipitation global precipitation estimates (i.e., IMERG-Late, GSMaP-

MVK, and PERSIANN-CCS) at a 0.5° spatial and daily temporal resolution in the JJA

season (Jun - Aug) across global land for the period from 2015 to 2019 (5 years).

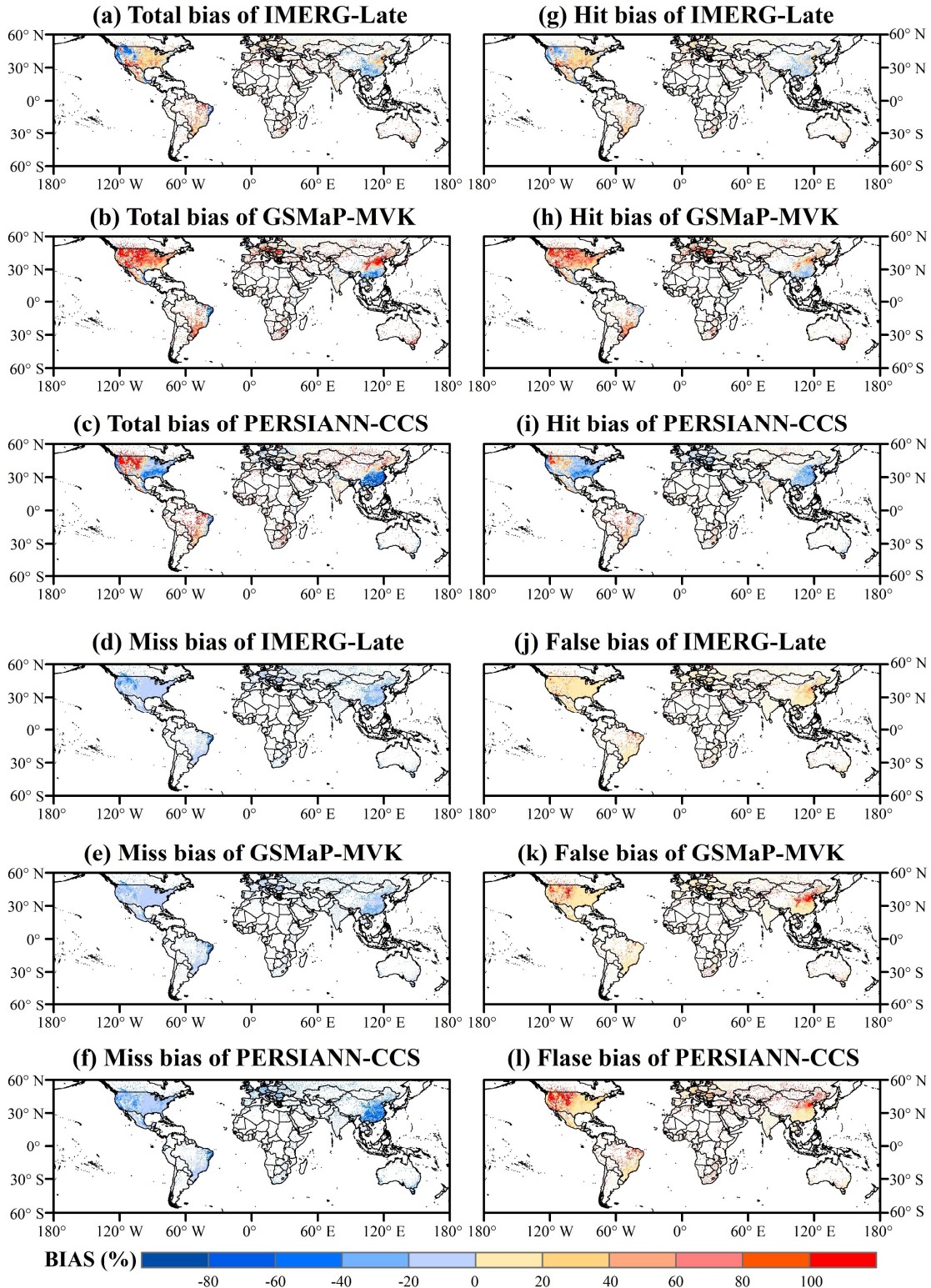

**Fig. 4.** Global maps of the total bias and its three independent components for the three

satellite-only precipitation global precipitation estimates (i.e., IMERG-Late, GSMaP-

MVK, and PERSIANN-CCS) at a 0.5° spatial and daily temporal resolution in the SON

season (Sep-Nov) across global land for the period from 2015 to 2019 (5 years).

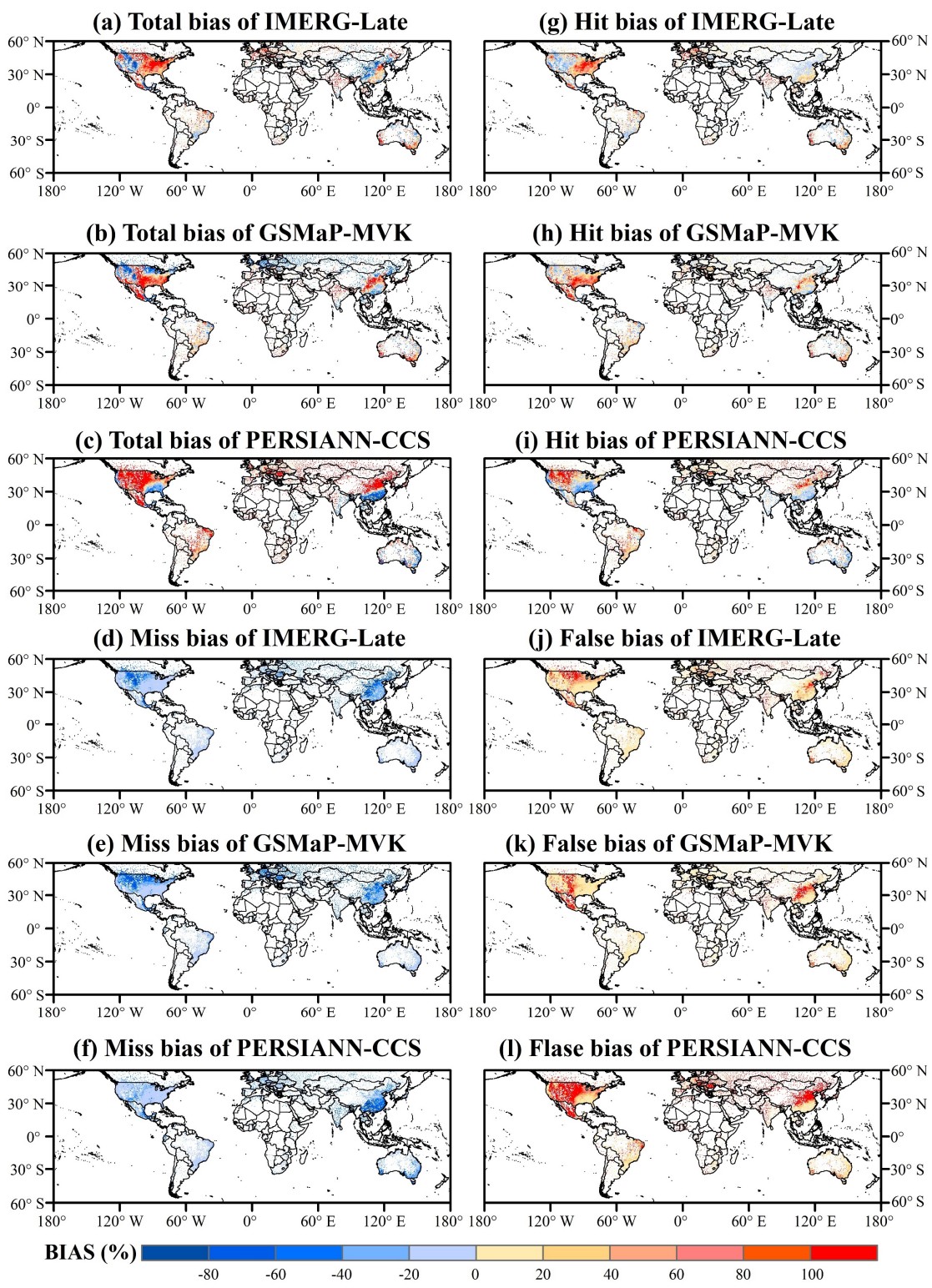


**Fig. 5.** Global maps of the total bias and its three independent components for the three

satellite-only precipitation global precipitation estimates (i.e., IMERG-Late, GSMaP-

MVK, and PERSIANN-CCS) at a 0.5° spatial and daily temporal resolution in the DJF

season (Dec-Feb) across global land for the period from 2015 to 2019 (5 years).


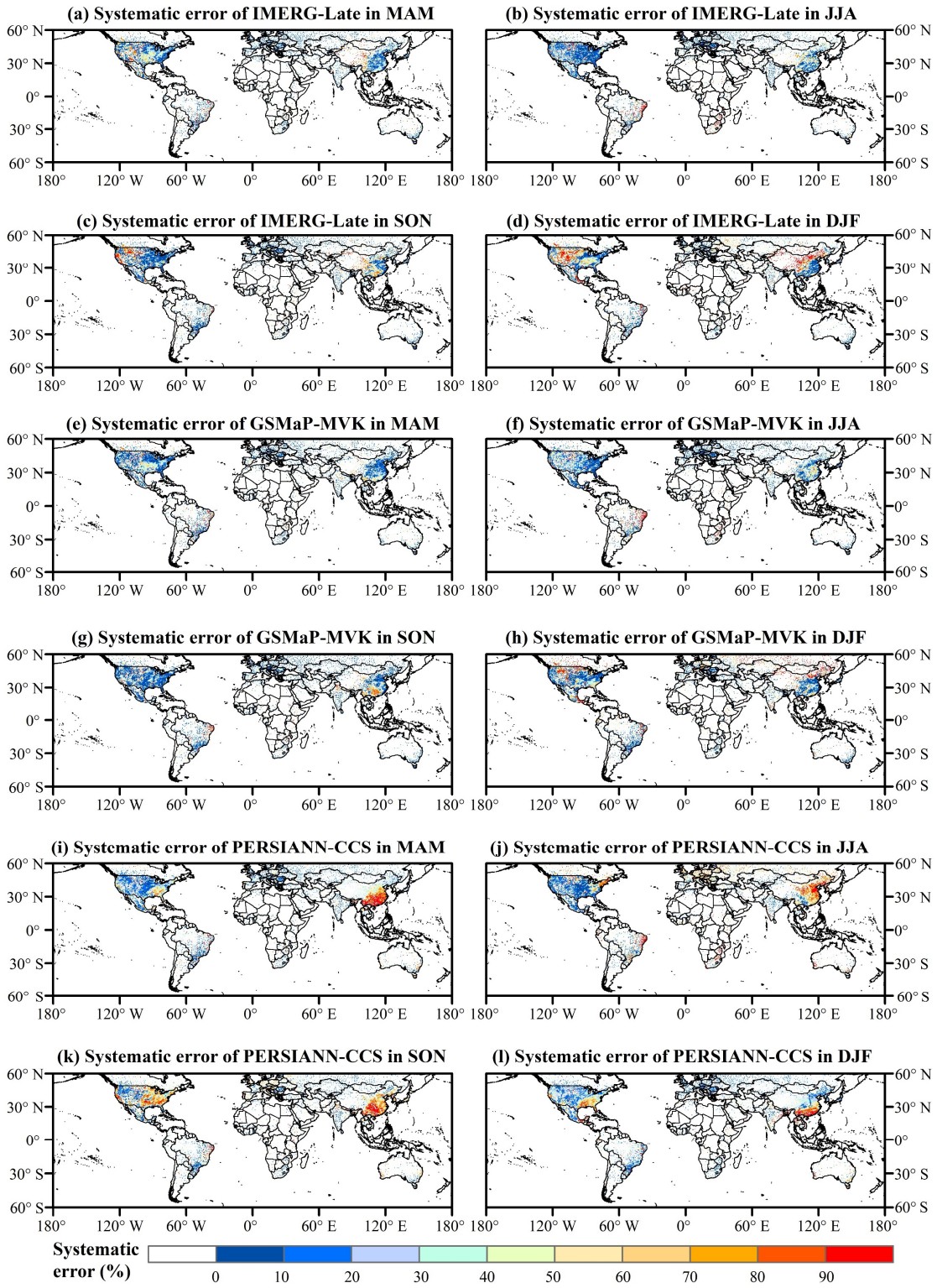

**Fig. 6.** Global maps of the systematic errors for the three satellite-only global precipitation estimates (i.e., IMERG-Late, GSMaP-MVK, and PERSIANN-CCS) at a

0.5° spatial and daily temporal resolution in four seasons (i.e., MAM, JJA, SON, and

DJF) over global land for the period from 2015 to 2019 (5 years): (a-d) IMERG-Late,

(e-h) GSMaP-MVK, (i-l) PERSIANN-CCS.

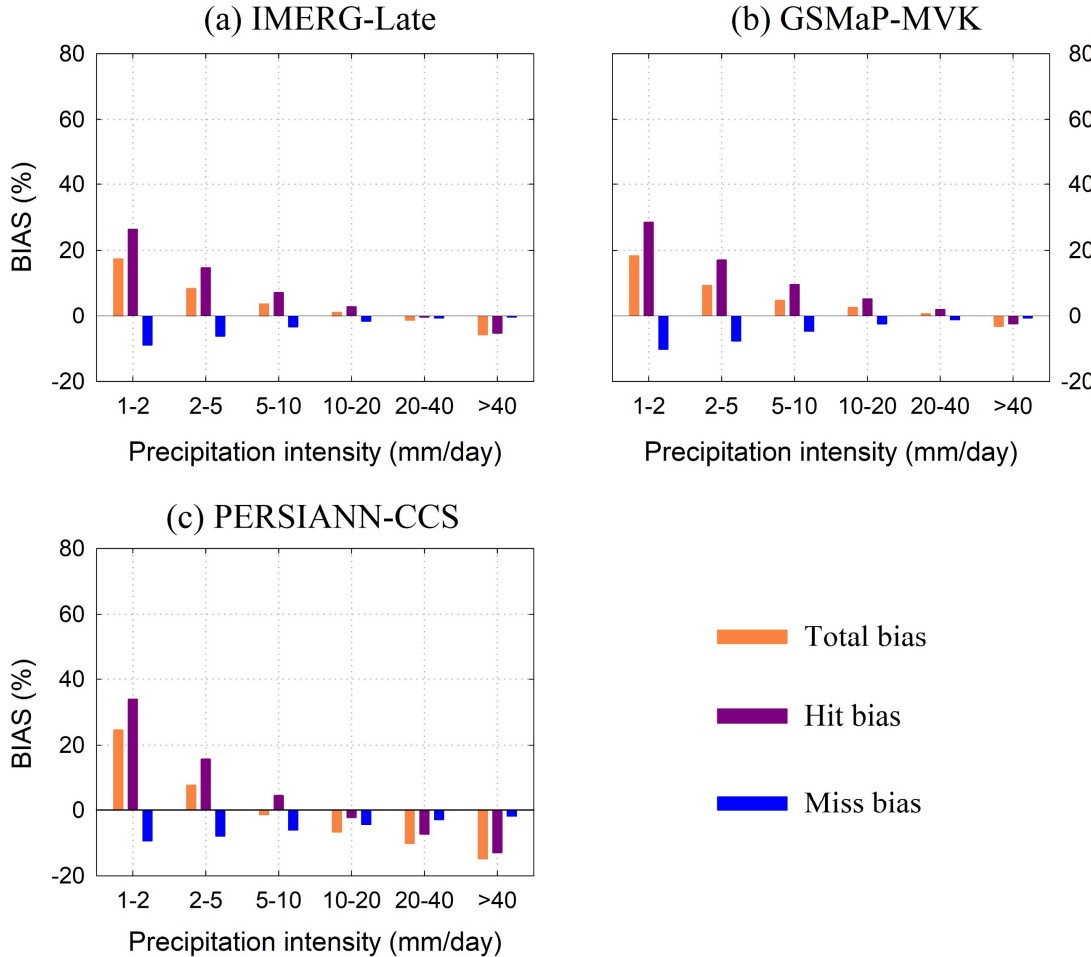

Fig. 7. Three bias scores (i.e., total bias, hit bias, and miss bias) of the three satellite-only global precipitation estimates (i.e., IMERG-Late, GSMaP-MVK, and PERSIANN-CCS) at a 0.5° spatial and daily temporal resolution under different rainfall intensities. Note that the precipitation intensity categories are from references.

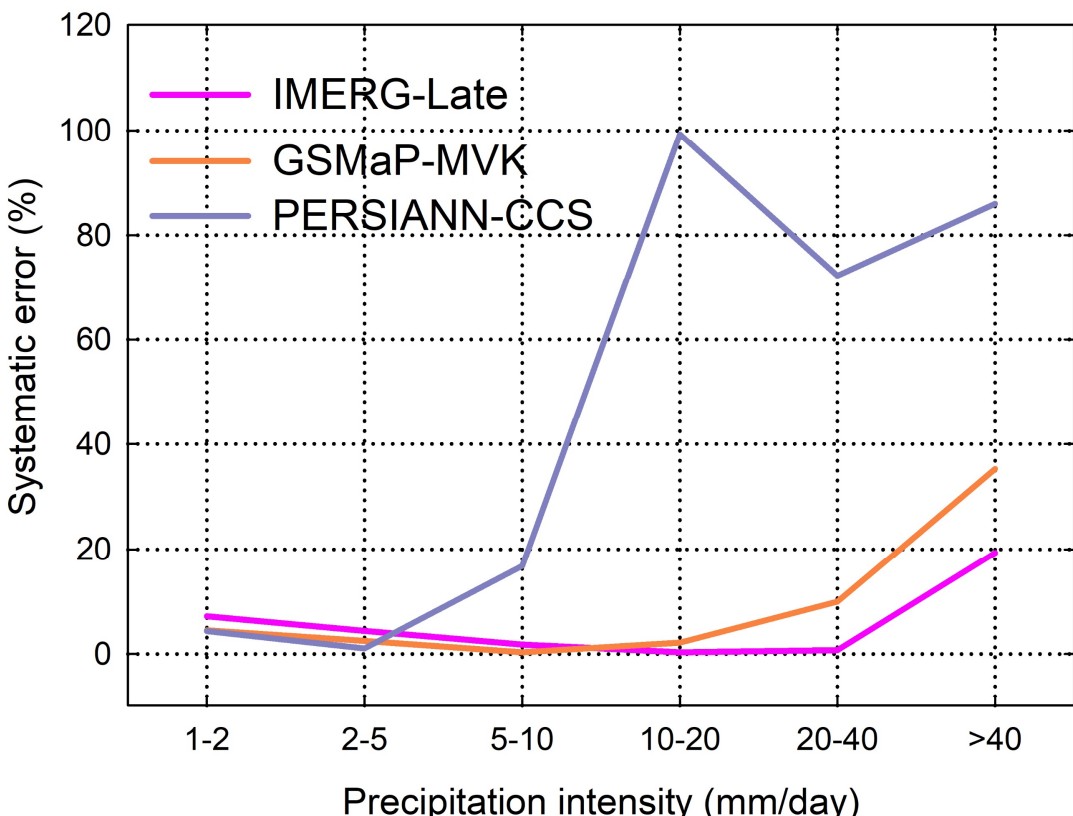

**Fig. 8.** Systematic errors of the three satellite-only global precipitation estimates (i.e., IMERG-Late, GSMaP-MVK, and PERSIANN-CCS) at a 0.5° spatial and daily temporal resolution under different rainfall intensities. Note that the precipitation intensity categories are from references.

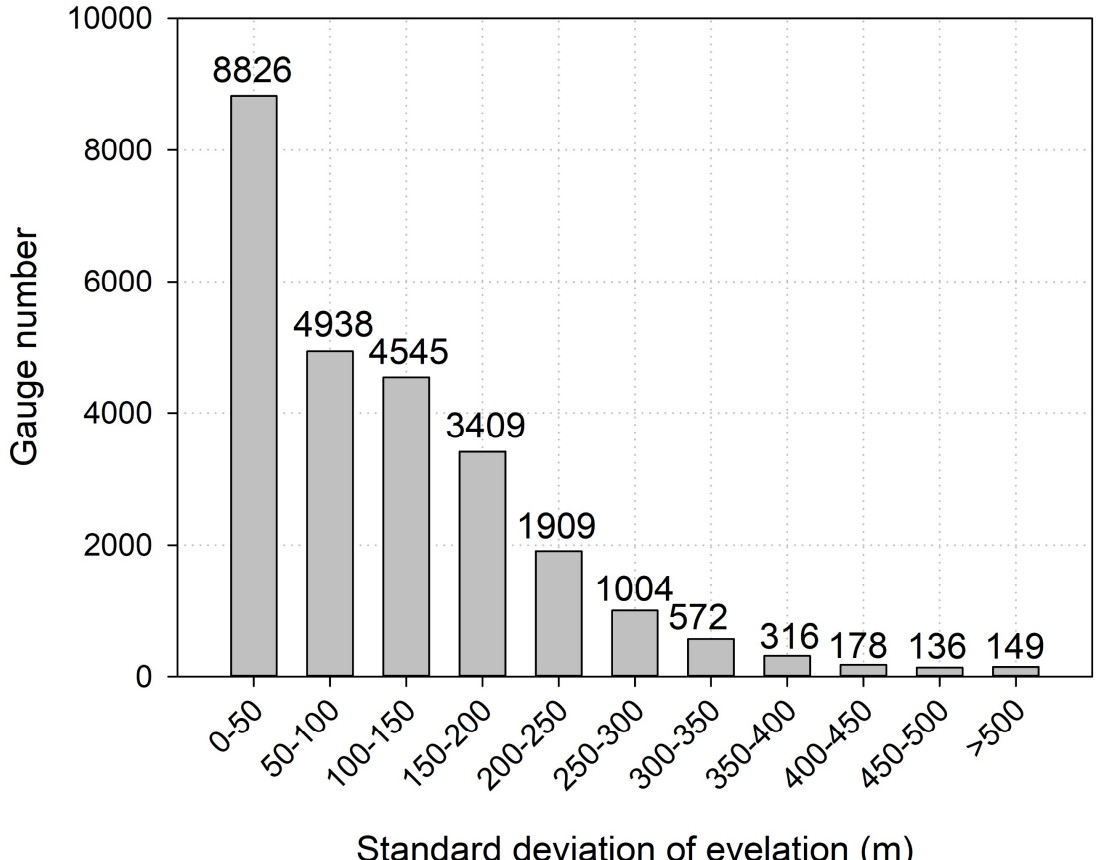


**Fig. 9**. Gauge number for each topography class.

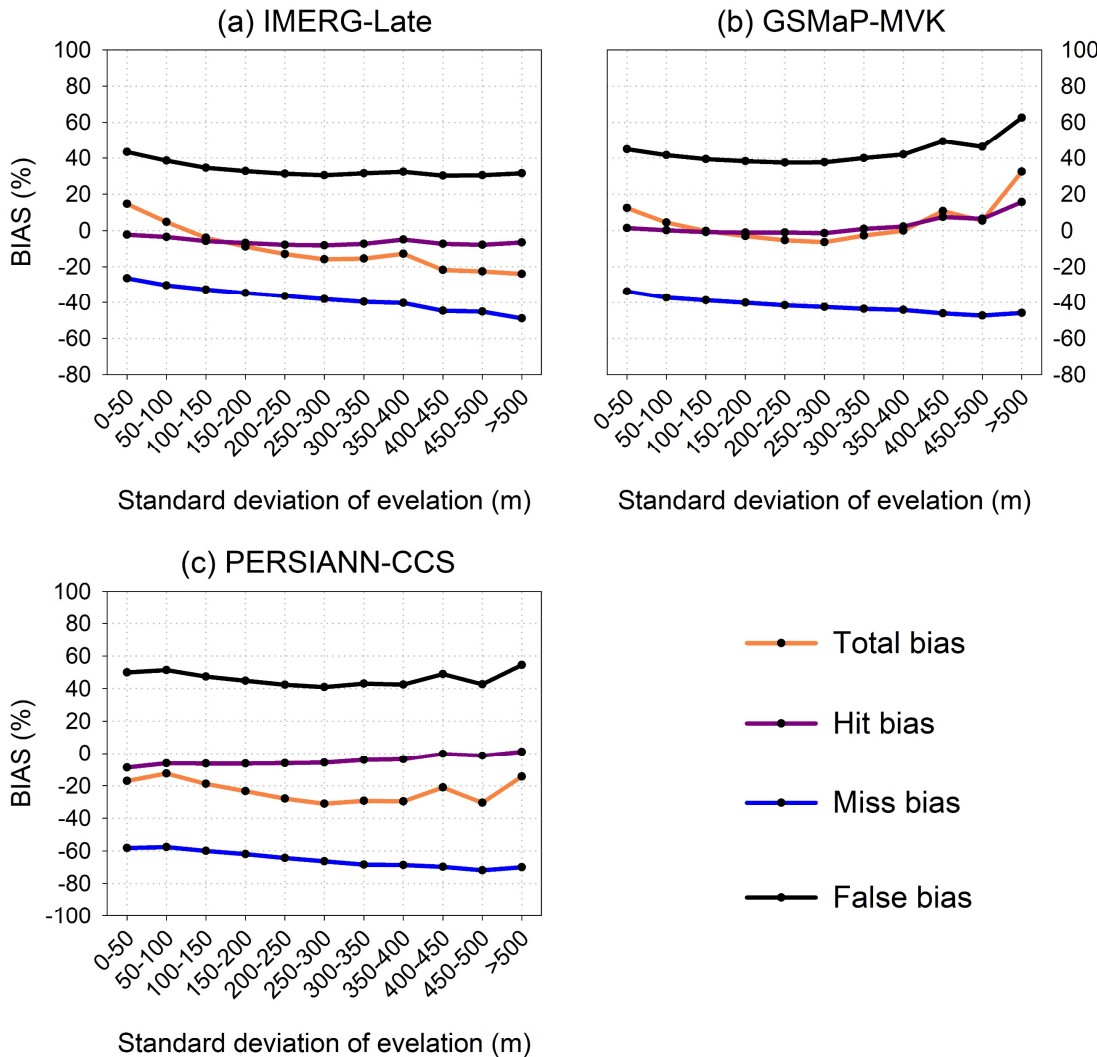

**Fig. 10.** Four bias scores (i.e., total bias, hit bias, miss bias, and false bias) of the three satellite-only global precipitation estimates (i.e., IMERG-Late, GSMaP-MVK, and PERSIANN-CCS) under different terrains. Note that the analysis executed at a 0.1° spatial and hourly temporal resolution in the humid regions of mainland China over JJA (summer) season for the period from 2015 to 2019 (5 years).


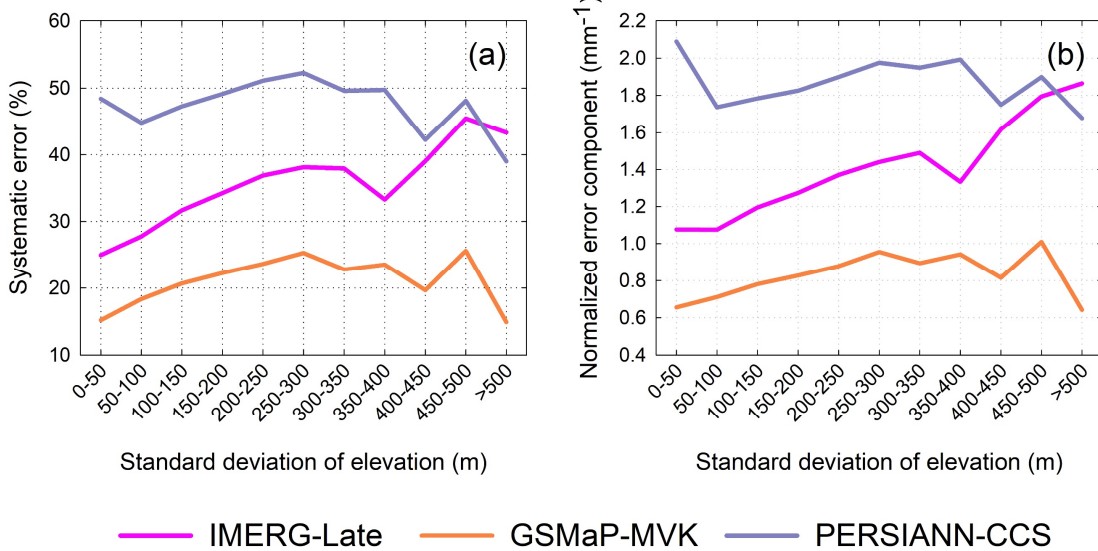

**Fig. 11.** (a) Systematic errors of the three satellite-only global precipitation estimates (i.e., IMERG-Late, GSMaP-MVK, and PERSIANN-CCS) under different topographies; (b) the variations of normalized error component (NEC) for the three satellite-only global precipitation estimates with increasing terrain complexity. Note that the analysis executed at a 0.1° spatial and hourly temporal resolution in the humid regions of mainland China over JJA (summer) season for the period from 2015 to 2019 (5 years).


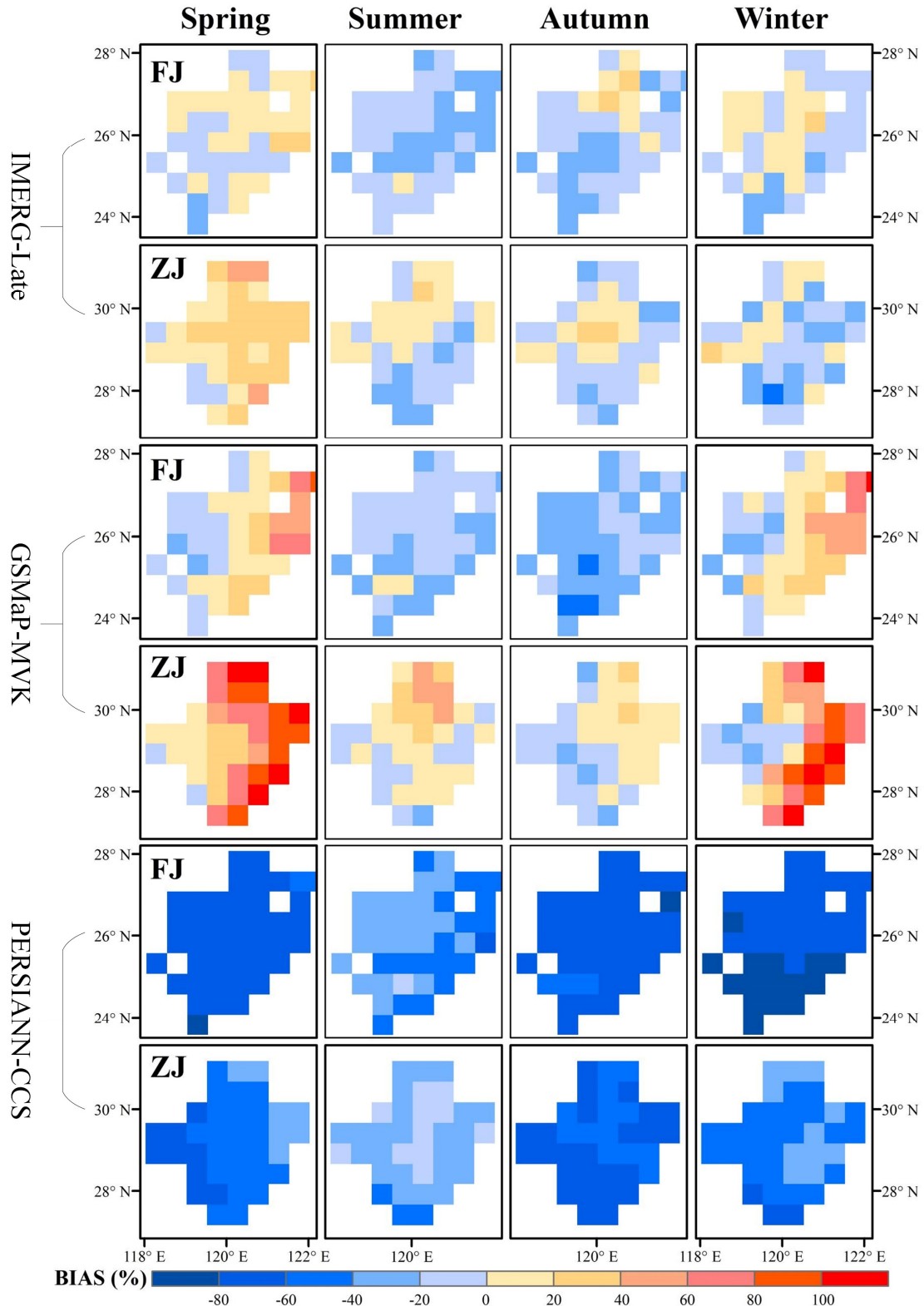

 **Fig. 12**. Spatial maps of the total biases of the three SPPs for four seasons over the

Fujian (FJ) and Zhejiang (ZJ) provinces, respectively.

**Table 1** The information about three satellite-only global precipitation estimates used in this study.

| Product | Full name of product | Data source | Resolution | Reference (s) |
|---|---|---|---|---|
| IMERG-Late | Integrated Multi-satellitE Retrievals for GPM Late run V06B | PMW, IR | 0.1°/0.5h | Huffman et al. (2019) |
| GSMaP-MVK | Global Satellite Mapping of Precipitation Microwave-IR Combined Product V7 | PMW, IR | 0.1°/1h | Ushio et al. (2009) |
| PERSIANN-CCS | Precipitation Estimation from Remotely Sensed Information using Artificial Neural Networks Cloud Classification | IR | 0.04°/1h | Sorooshian et al. (2000); Hong et al. (2004) |