# Peer review of "Global component analysis of errors in three satellite-only global precipitation estimates"

_Hydrology and Earth System Sciences, 2020_

## Referee Comment (RC1) · Anonymous Referee #1 · 15 Oct 2020

Understanding the error components of satellite products is very important to the algorithm developers and end-users. This manuscript presents the error analysis of five commonly used products. The finding is useful for the development on the future satellite retrieval algorithms. The topic is attractive and suitable for HESS. The detailed comments are as follows:

Major comments: 1. Given that the performance of IMERG-late is very similar to the performance of IMERG-early, I suggest removing one of them. 2. The dataset for China and for the rest of the world are different. I suggest authors discuss the effect of this inconsistency on the conclusion. 3. Section 4.1 Potential for the transferability of the regional assessment results to other areas. I don't think that comparison between CONUS and China is a good example to show the transferability. Although these two

regions have similarities in coverage area, and latitude range, the topography characteristics and climate regimes are very different. In addition, these two regions are very large and have dramatic heterogeneity in land surface property and climate, which is not suitable for the comparison. 4. Figure 9: What is the sample size of each category? What is the uncertainty of these results? 5. Figure 10 shows that the relation between the normalized error component (NEC) and elevation is very similar to that between the system error and elevation. What is the added value of NEC? What is significance of the index? Please clarify this.

Minor comments: 1. Line 17: IMERG-late, not MERG-late 2. Line 333: "false bias" should be "miss bias". 3. Lines 365: "increase" should be "decrease". 4. Figure 1: add the unit for density of rain gauge, add the source of precipitation data (1891-2018), and delete the legend on distance. 5. Figures 2-5: the objective of this study is to compare bias among different products. It could be more visually distinct to put the same bias category (for example, total bias) of different products in a column or row. 6. Figure 7: delete "false bias" in the description, as no false bias is presented in the figure.

―――――――――――――――――

---

## Referee Comment (RC2) · Anonymous Referee #2 · 2 Nov 2020

Although many publications already exist in the literature, errors and uncertainties associated with global satellite precipitation products are still difficult to characterize. Thus, I believe this work could be of interest to the HESS readership and worth publication. However, I have a few comments that I would like the authors to consider before accepting the manuscript for publication.

First off, I recommend revising the language, since there are a few grammatical mistakes.

Second, I suggest clarifying the main goal of this work. There are currently 5 goals mentioned at the end of the Introduction that read more like tasks. I think that a more focused article would be more effective. In other words, is the goal to validate the SPPs? Is it to model their errors/uncertainties? Is it to investigate what are the factors

causing more errors/uncertainty in one location/product/season vs. another? Is it to inform users and algorithm developers on how to use/improve such products?

In this regard, the abstract should be more concise and highlight the main goals and findings of this work.

Is including both IMERG Early and Late really necessary? The algorithm is same and – as expected – their performance very similar. Same goes for GSMaP-NRT and GSMaP-MVK. This may help with my comment above of a more focused article.

The section on "transferability of the regional assessment results to other areas" is weak and not well justified. However, this part of the study is also one of the most interesting, since improving our knowledge of how SPPs perform in regions of the world where no ground observations are available can be extremely useful (e.g., hydrologic applications). Ground observations are mainly available in plain areas (and sparse vegetation). Thus, how can we generalize such results to densely vegetated and highly complex regions? It would be useful to see how many gauges are available for each st dev class shown in Fig. 10.

---

## Author Comment (AC1) · 29 Nov 2020

Discussion comment 1 (The PDF version of the response can be found in supplement)

Understanding the error components of satellite products is very important to the algorithm developers and end-users. This manuscript presents the error analysis of five commonly used products. The finding is useful for the development on the future satellite retrieval algorithms. The topic is attractive and suitable for HESS.

We would like to sincerely thank the reviewer for his/her valuable comments and suggestions.

Major comments

[Figure]

Given that the performance of IMERG-late is very similar to the performance of IMERG-early, I suggest removing one of them.

Thanks. We will remove the IMERG-Early and GSMaP-NRT in revised version considering their similar performance relative to corresponding late-level products.

The dataset for China and for the rest of the world are different. I suggest authors discuss the effect of this inconsistency on the conclusion.

Good suggestion. In this study, these two different reference datasets were used to explore the error components of SPPs. The calculation processes of spatial maps of their biases and systematic errors include two major steps: (1)The spatial maps of biases and systematic error for the global land areas were given by using CPCU data as the benchmark, and then the ones of biases and systematic error for mainland China were given by using CGDPA data as the reference. (2)The spatial maps of biases and systematic error for mainland China using CGDPA data as the benchmark were used to replace the Chinese mainland part of the global land areas.

As for the error components of SPPs under different precipitation intensities, the precipitation values from satellite and ground reference over the globe (except for mainland China) were stored in vector S1 and vector G1, respectively. Then, the precipitation values from satellite and ground reference over mainland China were stored in vector S2 and vector G2, respectively. Third, S1 and S2 form a new vector S (S=[S1,S2]^T), while G1 and G2 form a new vector G (G=[G1,G2]^T). Finally, the error components of SPPs under different precipitation intensities were computed by using equations (1)-(9). Therefore, we can conclude that the inconsistency of reference data has little impact on the evaluation results. We will further clarify this point in the revision.

Section 4.1 Potential for the transferability of the regional assessment results to other areas. I don't think that comparison between CONUS and China is a good example to show the transferability. Although these two regions have similarities in coverage area, and latitude range, the topography characteristics and climate regimes are very

different. In addition, these two regions are very large and have dramatic heterogeneity in land surface property and climate, which is not suitable for the comparison

We thank the reviewer to propose this question. We fully agree with the opinion that the topography characteristics and climate regimes are rather different over COUNS and China. We also consider that these two areas might not be suitable to be used for discussing the transferability. In our revision, we have selected Chinese Fujian (FJ) and Zhejiang (ZJ) provinces to intercompare and analyze the transferability of evaluation results, due to these two adjacent areas located in southeastern China having similar topography and climate. The spatial maps of the total bias for the three SPPs over FJ and ZJ were shown in the following Fig. 1. One can see that the spatial maps of the total biases of each SPP between FJ and ZJ regions show significant differences for all four seasons. This indicates that the evaluation results between similar areas could not be extended to one another, which might be due to the large performance differences between various satellite sensors and the differences of the satellite samples used in satellite precipitation retrieval systems between various areas. We will revise this issue in the revised version.

Figure 9: What is the sample size of each category? What is the uncertainty of these results?

Thanks. We have provided the available gauges of each category, as shown in the following figure 2. One can see that the gauge number of each category is over 135, suggesting that the enough sample size of each class supports the reliability and robustness of the results. We will add the related discussion for this point in the section 3.3.

Figure 10 shows that the relation between the normalized error component (NEC) and elevation is very similar to that between the system error and elevation. What is the added value of NEC? What is significance of the index? Please clarify this.

Good questions. The relationship between the normalized error component (NEC) and

topography is very similar to that between the system error and topography, which is due to the close values of mean precipitation in all topography categories. This phenomenon can be explained by following definition equation of NEC (also see equation (10) in the manuscript) NEC=$(\sum\_(i = 1)\Theta n(S - G)\Theta2)/((\sum\_(i = 1)\Theta n(S - G)\Theta2)\times G\ \grave{I}\check{E}\ )$ The high similarity does not mean that NEC score is meaningless in investigating the impact of solely topographic factor on systematic error. This only indicates that when the mean precipitation of all terrain classes is very close, the relationship between NEC and topography degenerates into the relationship between systematic error and terrain. In practice, one of the functions of NEC is to exclude the interferences of the differences of precipitation intensity on systematic error, and it works only when the rainfall intensities are obviously different. However, the mean precipitation of all topography categories is very close in the humid regions of China, which leads to high similarity between the two relationship plots (i.e., Figs. 10a and b). In addition, NEC is a useful evaluation score in some cases, for example, avoiding increased values of systematic error with increasing temporal scales when investigating the relationship between systematic error and temporal scale. We will clarify the significance of this metric in section 2.3.2.

Minor comments

Line 17: IMERG-late, not MERG-late

Thanks, we have revised this mistake.

Line 333: "false bias" should be "miss bias".

Thanks, we have changed 'false bias' to 'miss bias'.

Lines 365: "increase" should be "decrease".

Thanks, we have replaced the word 'increase' with 'decrease'.

Figure 1: add the unit for density of rain gauge, add the source of precipitation data (1891-2018), and delete the legend on distance.

Ok, we will revise the Figure 1 according to this comment. The source of precipitation data (1891-2018) will be provide in the description.

Figures 2-5: the objective of this study is to compare bias among different products. It could be more visually distinct to put the same bias category (for example, total bias) of different products in a column or row.

We have completely revised the Figures 2-5 according to the reviewer's suggestion.

Figure 7: delete "false bias" in the description, as no false bias is presented in the figure.

OK, we have deleted 'false bias' in the description.

Please also note the supplement to this comment:
https://hess.copernicus.org/preprints/hess-2020-294/hess-2020-294-AC1-supplement.pdf
* * *
[Figure]

**Fig. 1.** The spatial maps of the total biases of three SPPs for four seasons over the Fujian (FJ) and Zhejiang (ZJ) provinces, respectively.

The gauge number of each topography class chart:

8826 — 0-50
4938 — 50-100
4545 — 100-150
3409 — 150-200
1909 — 200-250
1004 — 250-300
572 — 300-350
316 — 350-400
178 — 400-450
136 — 450-500
149 — >500

*Y-axis:* Gauge number (0 to 10000)
*X-axis:* Standard deviation of evelation (m)

**Fig. 2.** The gauge number of each topography class.

---

## Author Comment (AC2) · 29 Nov 2020

Discussion comment 2 (The PDF version of the response can be found in supplement)

Although many publications already exist in the literature, errors and uncertainties associated with global satellite precipitation products are still difficult to characterize. Thus, I believe this work could be of interest to the HESS readership and worth publication. However, I have a few comments that I would like the authors to consider before accepting the manuscript for publication.

Many thanks for the important comments and suggestions from this reviewer.

1. First off, I recommend revising the language, since there are a few grammatical mistakes

[Figure]

In the revision, the manuscript will be carefully checked and revised to avoid all the grammatical mistakes.

2. Second, I suggest clarifying the main goal of this work. There are currently 5 goals mentioned at the end of the Introduction that read more like tasks. I think that a more focused article would be more effective. In other words, is the goal to validate the SPPs? Is it to model their errors/uncertainties? Is it to investigate what are the factors causing more errors/uncertainty in one location/product/season vs. another? Is it to inform users and algorithm developers on how to use/improve such products?

Thank you for the suggestions. We will further clarify the main goal of this work at the end of the Introduction according to reviewer's suggestion.

3. In this regard, the abstract should be more concise and highlight the main goals and findings of this work

Ok, we will further refine the abstract to highlight the main goals and findings of this work.

4. Is including both IMERG Early and Late really necessary? The algorithm is same and – as expected – their performance very similar. Same goes for GSMaP-NRT and GSMaP-MVK. This may help with my comment above of a more focused article.

Thank you. In fact, the performance of IMERG-Early is similar to that of IMERG-Late. According to reviewer's suggestion, we will remove the IMERG-Early and GSMaP-NRT in our revised manuscript.

5. The section on "transferability of the regional assessment results to other areas" is weak and not well justified. However, this part of the study is also one of the most interesting, since improving our knowledge of how SPPs perform in regions of the world where no ground observations are available can be extremely useful (e.g., hydrologic applications). Ground observations are mainly available in plain areas (and sparse vegetation). Thus, how can we generalize such results to densely vegetated and highly

complex regions?

Thanks. We entirely agree with the reviewer's opinion. The verification about the transferability of the evaluation results seems weak and is not well justified. However, this part of the study is rather important. Therefore, in the revision we chose two suitable areas (i.e., Chinese Fujian (FJ) and Zhejiang (ZJ) provinces) for comparing and investigating the potential for the transferability of the regional assessment results to other areas. These two selected adjacent areas, which located in southeastern China, have similar topography and climate. The spatial distribution of the total bias for each SPP over FJ and ZJ were given in the following Fig.1. Apparently, the spatial distributions of the total biases between FJ and ZJ exhibit evident differences, suggesting that the evaluation results between similar areas could not be extended to one another. The reason might be attributed to the large performance differences existed between various satellite sensors and the differences of the satellite samples in retrieval systems over different regions. We will revise this issue in this revised version.

6. It would be useful to see how many gauges are available for each stdev class shown in Fig. 10.

Ok, we will provide the available gauges for each topography class in the version, as shown in the following Fig.2. One can see that enough ground rain gauges, exceeding 135 rain gauges for each terrain class, were used as the references to ensure the reliability and robustness of evaluation results. We will add the corresponding discussion for this point in the section 3.3.

Please also note the supplement to this comment:
https://hess.copernicus.org/preprints/hess-2020-294/hess-2020-294-AC2-supplement.pdf
* * *
[Figure]

**Fig. 1.** The spatial maps of the total biases of three SPPs for four seasons over the Fujian (FJ) and Zhejiang (ZJ) provinces, respectively.

Gauge number

8826

4938
4545

3409

1909

1004
572

316 178 136 149

0-50 50-100 100-150 150-200 200-250 250-300 300-350 350-400 400-450 450-500 >500

Standard deviation of evelation (m)

**Fig. 2.** Fig. 2 The gauge number of each topography class.

---

## Author Response (AR1)

**Response to review on manuscript: [hess-2020-294]**

**Previous title:** Global component analysis of errors in five satellite-only global precipitation estimates

**Current title:** Global component analysis of errors in three satellite-only global precipitation estimates

**Journal:** Hydrology and Earth System Sciences

**Corresponding author:** Bin Yong

**Authors:** Hanqing Chen, Bin Yong[*], Pierre-Emmanuel Kirstetter, Leyang Wang, Yang Hong

Dear Editor, Graham Jewitt:

We would like to thank you for giving us the opportunity to revise our paper (**Major Revision**) and thank two anonymous reviewers for their careful reviews and their valuable comments. In the revision, we have addressed all the requests and comments of the reviewers point by point. For convenience, the responses to the comments were written down below in blue font. The modifications made in the revised text were highlighted in red font. We believe that the current version is suitable for publication in Hydrology and Earth System Sciences.

If you have any questions about this manuscript, please do not hesitate to let me know.

Sincerely yours,

Dr. Bin Yong

Collaborative Innovation Center on Forecast and Evaluation of Meteorological Disasters, Institute for Disaster Risk Management, School of Geographical Science, Nanjing University of Information Science and Technology, Nanjing 210000, China.

Email: yongbin_hhu@126.com

**Response for Reviewer #1**

**Understanding the error components of satellite products is very important to the algorithm developers and end-users. This manuscript presents the error analysis of five commonly used products. The finding is useful for the development on the future satellite retrieval algorithms. The topic is attractive and suitable for HESS.**

**Response:** We would like to sincerely thank the reviewer for his/her valuable comments and suggestions.

**Major comments**

1. **Given that the performance of IMERG-late is very similar to the performance of IMERG-early, I suggest removing one of them.**

**Response:** Thanks. We have removed the IMERG-Early and GSMaP-NRT in the revised version considering their similar performance relative to their corresponding late-level products.

2. **The dataset for China and for the rest of the world are different. I suggest authors discuss the effect of this inconsistency on the conclusion.**

**Response:** Good suggestion. In this study, these two different reference datasets were used to explore the error components of SPPs. The calculation processes of the spatial maps of biases and systematic errors include two major steps:

**(1) Global maps of biases and systematic error**

   a. The spatial maps of biases and systematic error computed over the global land areas use CPCU data as the benchmark. Maps of biases and systematic error over mainland China use CGDPA data as the reference.

   b. The spatial maps of biases and systematic error over mainland China replace the Chinese mainland part of the global land areas.

**(2) Error components of SPPs under different precipitation intensities**

The satellite and ground reference precipitation values over the globe (except for mainland China) were stored in vector S1 and vector G1, respectively. Then the satellite and ground reference precipitation values over mainland China were stored in vector S2 and vector G2, respectively. Third, S1 and S2 form a new vector S (S $= [S1, S2]^T$), while G1 and G2 form a new vector G (G $= [G1, G2]^T$). Finally, the SPP error components under different precipitation intensities were computed by using equations (1)-(9).

The inconsistency of reference data has little impact on the evaluation results. We have discussed this in **lines 141-144**. The calculation processes were provided as

supplementary materials, **see supplementary material**.

3. **Section 4.1 Potential for the transferability of the regional assessment results to other areas. I don't think that comparison between CONUS and China is a good example to show the transferability. Although these two regions have similarities in coverage area, and latitude range, the topography characteristics and climate regimes are very different. In addition, these two regions are very large and have dramatic heterogeneity in land surface property and climate, which is not suitable for the comparison**

**Response:** We thank the reviewer for this comment. We agree that the characteristics in terms of topography and climate regimes are different over CONUS and China, making them challenging for discussing the transferability. In our revision, we have selected the Chinese Fujian (FJ) and Zhejiang (ZJ) provinces to intercompare and analyze the transferability of evaluation results. These two adjacent areas are located in southeastern China and having similar topography and climate. Spatial maps of the total bias for the three SPPs over FJ and ZJ are shown in the following Fig. 1. One can see that the spatial maps of the total biases for each SPP show significant differences for all four seasons between the FJ and ZJ regions. Besides, we found that the total biases in the FJ (or ZJ) areas show evident differences. This indicates that the evaluation results between these similar areas cannot be extended to one another. It might be due to the large differences in performance between the various satellite sensors and differences in satellite samples used in the satellite precipitation retrieval systems between the areas. Also, it might be also caused by factors (e.g. characteristics of

precipitation regimes, such as precipitation types) not captured by satellites or the reference. We have discussed this issue in the revised version, **see section 4.1**.

[Figure]

Fig. 1 Spatial maps of total bias for the three SPPs for four seasons over the Fujian (FJ) and Zhejiang (ZJ) provinces, respectively.

4. **Figure 9: What is the sample size of each category? What is the uncertainty of these results?**

**Response:** Thanks. We provided the gauges available for each category as shown in the following Figure 2. One can see that the gauge number in each category is over 135,

which suggests that the sample size in each class is sufficient to support the reliability and robustness of the results. We have added the related descriptions in section 3.3, **see lines 319-321**.

[Figure]

Fig. 2 Gauge number for each topography class.

5. **Figure 10 shows that the relation between the normalized error component (NEC) and elevation is very similar to that between the system error and elevation. What is the added value of NEC? What is significance of the index? Please clarify this.**

**Response:** Good questions. The relationship between the normalized error component (NEC) and topography is very similar to that between the system error and topography.

It is due to close values of mean precipitation in all topography categories. This phenomenon can be explained by the following equation for NEC (also see equation (10) in the manuscript)

$$NEC = \left(\sum_{i=1}^{n}(\hat{S} - G)^2\right) / \left((\sum_{i=1}^{n}(S - G)^2) \times \bar{G}\right)$$

The relationship between NEC and topography degenerates into the relationship between systematic error and topography when the mean precipitation (i.e., $\bar{G}$) of all topography classes is very close. However, the high similarity does not mean that the NEC score is meaningless for investigating the impact of topography on the systematic error. In practice, the purpose of the NEC is to exclude the impact of other factors such as precipitation intensity on the systematic error. It works only when the rainfall intensities are significantly different. We have clarified the significance of this metric in section 2.3.2, **see lines 211-214**.

**Minor comments**

1. **Line 17: IMERG-late, not MERG-late**

**Response:** Thanks, we have revised this mistake, **see lines 20**.

2. **Line 333: "false bias" should be "miss bias".**

**Response:** Thanks, we have changed 'false bias' to 'miss bias', **see line 299**.

3. **Lines 365: "increase" should be "decrease".**

**Response:** Thanks, we have replaced the word 'increase' with 'decrease', **see line 325**.

4. **Figure 1: add the unit for density of rain gauge, add the source of precipitation data (1891-2018), and delete the legend on distance.**

**Response:** We have revised the Figure 1 according to this comment. The source of precipitation data (1891-2018) has provided in the description, **see Fig.1 and lines 689-690**.

5. **Figures 2-5: the objective of this study is to compare bias among different products. It could be more visually distinct to put the same bias category (for example, total bias) of different products in a column or row.**

**Response:** We have completely revised the Figures 2-5 according to the reviewer's suggestion, **see Figs. 2-5 in the revised version**.

6. **Figure 7: delete "false bias" in the description, as no false bias is presented in the figure.**

**Response:** We have deleted 'false bias' in the description, **see line 717**.

**In addition, we did the experiment again because we found that the previous results in mainland China were from mismatched satellite-ground observations (CGDPA) pairs, and all figures and tables have been revised. Other revisions were highlighted in red font.**

**Thanks for your review, we look forward to your positive response.**

**Response for reviewer #2**

**Although many publications already exist in the literature, errors and uncertainties associated with global satellite precipitation products are still difficult to characterize. Thus, I believe this work could be of interest to the HESS readership and worth publication. However, I have a few comments that I would like the authors to consider before accepting the manuscript for publication.**

**Response:** Many thanks for the important comments and suggestions from this reviewer.

1. **First off, I recommend revising the language, since there are a few grammatical mistakes**

**Response:** The manuscript has been carefully checked and revised to avoid all grammatical mistakes.

2. **Second, I suggest clarifying the main goal of this work. There are currently 5 goals mentioned at the end of the Introduction that read more like tasks. I think that a more focused article would be more effective. In other words, is the goal to validate the SPPs? Is it to model their errors/uncertainties? Is it to investigate what are the factors causing more errors/uncertainty in one location/product/season vs. another? Is it to inform users and algorithm developers on how to use/improve such products?**

**Response:** Thank you for the suggestions. We have further clarified the main goal of this work at the end of the Introduction. The revised content shows:

The objectives of this study are four-fold: (1) to reveal the major components of errors (including total bias and total mean squared error) for three SPPs including IMERG Late run (IMERG-Late), GSMaP Microwave-IR Combined Product (GSMaP-MVK), and PERSIANN-CCS for four seasons across global land areas; (2) to investigate the potential for the transferability of the regional assessment results to other similar regions; (3) to investigate what are the factors causing large systematic errors; (4) to inform users and algorithm developers on how to improve these satellite precipitation products.

The detailed revision can be found in **lines 109-116.**

3. **In this regard, the abstract should be more concise and highlight the main goals and findings of this work**

**Response:** We have further refined the abstract to highlight the main goals and findings of this work, **see Abstract**.

4. **Is including both IMERG Early and Late really necessary? The algorithm is same and – as expected – their performance very similar. Same goes for GSMaP-NRT and GSMaP-MVK. This may help with my comment above of a more focused article.**

**Response:** Thank you. In fact, the performance of IMERG-Early is similar to that of IMERG-Late. According to the reviewer's suggestion, we have removed the IMERG-Early and GSMaP-NRT in our revised manuscript.

5. **The section on "transferability of the regional assessment results to other areas" is weak and not well justified. However, this part of the study is also one of the most interesting, since improving our knowledge of how SPPs perform in regions of the world where no ground observations are available can be extremely useful (e.g., hydrologic applications). Ground observations are mainly available in plain areas (and sparse vegetation). Thus, how can we generalize such results to densely vegetated and highly complex regions?**

**Response:** Thanks. We entirely agree with the reviewer's opinion. The discussion on transferability of the evaluation results seems weak and is not well justified. However, this part of the study is rather important. Therefore, in the revision we chose two suitable areas (i.e., Chinese Fujian (FJ) and Zhejiang (ZJ) provinces) for comparing and investigating the potential for transferability of the regional assessment results to other areas. These two selected adjacent areas, which are located in southeastern China, have similar topography and climate. The spatial distribution of the total bias for each SPP over FJ and ZJ are given in the following Fig.1. The spatial distributions of the total biases exhibit significant differences between FJ and ZJ, suggesting that the evaluation results between similar areas cannot be extended to one another. This indicates that the evaluation results between these similar areas cannot be extended to one another. It might be due to the large differences in performance between the various satellite sensors and differences in satellite samples used in the satellite precipitation retrieval systems between the areas. Also, it might be also caused by factors (e.g. characteristics of precipitation regimes, such as precipitation types) not captured by satellites or the

reference. We have discussed this issue in the revised version, **see section 4.1**.

[Figure]

Fig. 1 The spatial maps of the total biases of three SPPs for four seasons over the Fujian

(FJ) and Zhejiang (ZJ) provinces, respectively.

**6. It would be useful to see how many gauges are available for each stdev class**

   **shown in Fig. 10.**

**Response:** Thanks. We provided the gauges available for each category as shown in

the following Figure 2. One can see that the gauge number in each category is over 135, which suggests that the sample size in each class is sufficient to support the reliability and robustness of the results. We have added the related descriptions in section 3.3, **see lines 319-321**.

[Figure]

Fig. 2 The gauge number of each topography class.

**In addition, we did the experiment again because we found that the previous results in mainland China were from mismatched satellite-ground observations (CGDPA) pairs, and all figures and tables have been revised. Other revisions were highlighted in red font.**

Thanks for your review, we look forward to your positive response.

---

## Author Response (AR3)

**Response to review on manuscript: [hess-2020-294]**

**Title:** Global component analysis of errors in three satellite-only global precipitation estimates

**Journal:** Hydrology and Earth System Sciences

**Corresponding author:** Bin Yong

**Authors:** Hanqing Chen, Bin Yong[*], Pierre-Emmanuel Kirstetter, Leyang Wang, Yang Hong

Dear Editor, Graham Jewitt:

We are glad to receive your positive comments on the paper. We would like to express our most sincere thanks to you and two reviewers. In the revision, we have addressed all the requests and comments of the editor and the reviewers point by point. For convenience, the responses to the comments were written down below in blue font. We believe that the current version is suitable for publication in Hydrology and Earth System Sciences.

If you have any questions about this manuscript, please do not hesitate to let me know.

Sincerely yours,

Dr. Bin Yong

State Key Laboratory of Hydrology-Water Resources and Hydraulic Engineering,

Hohai University, Nanjing 210098, China.

Email: yongbin_hhu@126.com

**Response for Editor**

Copy-editing of the manuscript will be done automatically, but please check that you are happy with the language used in teh context of the reviewers comments.

Finally, please consider the colours used in your figures. I think that red-green colour blind people will struggle with some of them.

In Figure 7, it is very difficult to distuingish between the colours used.

Response: We have modified this Figure.

In Figure 8, 10 and 11 colours are difficult to see and lines are very thin.

Response: We have modified the three Figures.

The editorial team will indicate whether they find your captions adequate. "As in Fig. x" is unlikely to be accepted - in general Captions should be able to stand alone from the paper they appear in.

Response: We have revised this issue, see Captions (pages 31-33).

**Review #1**

The authors did a great job with addressing my comments and I believe the manuscript is now suitable for publication. There are still a few minor grammatical errors that can

be easily fixed with a final revision.

For instance, it should be either "The objectives of this study are four" or "The objective of this study is four-fold".

Response: Thanks. We have revised this grammatical error (see line 106), and checked carefully the full text.